# Generation of a humanized Aβ expressing mouse demonstrating aspects of Alzheimer's disease-like pathology

David Baglietto-Vargas [1,2,3], Stefania Forner [1], Lena Cai[1], Alessandra C. Martini [1], Laura Trujillo-Estrada[1,3], Vivek Swarup [1,2], Marie Minh Thu Nguyen[1], Kelly Do Huynh[1], Dominic I. Javonillo[1], Kristine Minh Tran[1], Jimmy Phan[1], Shan Jiang[4], Enikö A. Kramár[1,2], Cristina Nuñez-Diaz[3], Gabriela Balderrama-Gutierrez[4], Franklin Garcia [1,2], Jessica Childs[1,2], Carlos J. Rodriguez-Ortiz [1,5], Juan Antonio Garcia-Leon[3], Masashi Kitazawa [1,5], Mohammad Shahnawaz[6], Dina P. Matheos[1,2], Xinyi Ma[4], Celia Da Cunha[1], Ken C. Walls[1], Rahasson R. Ager[1], Claudio Soto[6], Antonia Gutierrez [3], Ines Moreno-Gonzalez [3,6], Ali Mortazavi [4], Andrea J. Tenner [1,2,7], Grant R. MacGregor [4], Marcelo Wood[1,2], Kim N. Green [1,2 ✉] & Frank M. LaFerla [1,2 ✉]

The majority of Alzheimer's disease (AD) cases are late-onset and occur sporadically, however most mouse models of the disease harbor pathogenic mutations, rendering them better representations of familial autosomal-dominant forms of the disease. Here, we generated knock-in mice that express wildtype human Aβ under control of the mouse *App* locus. Remarkably, changing 3 amino acids in the mouse Aβ sequence to its wild-type human counterpart leads to age-dependent impairments in cognition and synaptic plasticity, brain volumetric changes, inflammatory alterations, the appearance of Periodic Acid-Schiff (PAS) granules and changes in gene expression. In addition, when exon 14 encoding the Aβ sequence was flanked by *loxP* sites we show that Cre-mediated excision of exon 14 ablates hAβ expression, rescues cognition and reduces the formation of PAS granules.

[1] Institute for Memory Impairments and Neurological Disorders, University of California, Irvine, CA, USA. [2] Department of Neurobiology and Behavior, University of California, Irvine, CA, USA. [3] Department of Cell Biology, Genetic and Physiology, Faculty of Sciences, Instituto de Investigacion Biomedica de Malaga-IBIMA, Networking Research Center on Neurodegenerative Diseases (CIBERNED), University of Malaga, Malaga, Spain. [4] Department of Developmental and Cell Biology, University of California, Irvine, CA, USA. [5] Division of Occupational and Environmental Medicine, Department of Medicine. Center for Occupational and Environmental Health (COEH), University of California, Irvine, CA, USA. [6] The Mitchell Center for Alzheimer's Disease and Related Brain Disorders, Department of Neurology, McGovern Medical School, University of Texas Health Science Center at Houston, Houston, TX, USA. [7] Department of Molecular Biology and Biochemistry, University of California, Irvine, CA, USA. ✉email: kngreen@uci.edu; laferla@uci.edu

The development of successful treatments for complex human disorders such as Alzheimer's disease (AD) can be facilitated by animal models that recapitulate key phenotypic features found in the human disease condition. To date, over 170 genetically modified mouse models containing AD-linked mutations have been generated to study this progressive neurodegenerative disorder, and in many instances these mice have yielded insights into the underlying pathogenic mechanisms[1–4]. However, therapeutic approaches that have been demonstrated to be successful in these familial AD (FAD) models have failed when evaluated in human clinical trials involving participants with late-onset AD (LOAD)[5–7]. As existing mouse models of AD frequently incorporate disease-causing mutations in one or more genes associated with autosomal-dominant dementias and because the gene products are often expressed at supra-physiological levels, there is a need to develop new animal models that better recapitulate the underlying molecular pathways leading to late-onset sporadic AD[6–10].

Towards this goal, we humanized the Aβ peptide-coding sequence in the mouse *amyloid precursor protein* (*App*) gene to generate a non-mutant human Aβ knock-in (hAβ-KI) mouse. This model has several key features: (1) no familial AD mutations are included and the mouse expresses the wild-type human sequence of the amyloid precursor protein (APP) cleavage product; (2) the endogenous gene-regulatory elements are used so APP is produced at murine physiological levels; and (3) *loxP* sites flank the exon encoding Aβ allowing for cell-specific/temporal control of Aβ/APP production to enable further cell-specific mechanistic investigation.

hAβ-KI mice develop age-dependent changes in behavior, synaptic plasticity, inflammatory response, the formation of Periodic Acid-Schiff (PAS) positive granules and the transcriptome (such as metabolic/energetic and neuroplasticity gene expression), that mimic the late-onset progression seen in sporadic human AD cases. Cognitive function in hAβ-KI mice could be restored following virally-mediated transduction of CRE to ablate Aβ production. In addition, the number of PAS granules were reduced in the hAβ-KI mice upon using the CRE system. This new mouse strain will serve as a useful platform to investigate the many genetic, aging, and environmental factors that drive the development of AD.

## Results

### Generation of a human Aβ knock-in mouse.
Homologous recombination in mouse embryonic stem (ES) cells was used to generate mice in which the murine *App* gene was humanized by changing 3 amino acids within the Aβ peptide sequence (hAβ-KI) (Fig. 1a, b). No autosomal-dominant APP mutation was introduced (Supplementary Fig. 1a). The resulting hAβ-KI allele expresses human wildtype (WT) Aβ (hAβ) in the context of the cognate *App* locus, and the regional localization and extent of APP expression appear indistinguishable from the unmodified allele. The hAβ-KI allele includes *loxP* sites flanking exon 14 encoding Aβ, allowing CRE recombinase-mediated cessation of Aβ production to assess the dependence of pathological sequelae on Aβ production (Fig. 1c). To verify the *loxP* sites were functional, we crossed hAβ-KI mice with mice expressing tamoxifen-inducible Cre recombinase under the control of the ubiquitin promoter (UBC-Cre[ERT2] mice). After induction of Cre expression with tamoxifen which should delete exon 14 from the floxed allele in the hAβ-KI heterozygous animals, immunoblot analysis demonstrated a reduction in APP quantity in brain relative to uninduced controls (Fig. 1d and Supplementary Fig. 8). Furthermore, humanized APP was completely suppressed in tamoxifen-treated hAβ-KI heterozygous; UBC-Cre[ERT2] hemizygous mice versus

vehicle-treated controls of the same genotype (Fig. 1d and Supplementary Fig. 8). Consistent with these results, real-time qPCR analysis indicated that *App* mRNA expression in tamoxifen-treated hAβ-KI heterozygous; UBC-Cre[ERT2] hemizygous mice was reduced compared to PBS-treated mice of the same genotype (Fig. 1e). The reduction in *App* (as well as Aβ) after CRE-mediated deletion of the exon encoding hAβ may be a consequence of mRNA instability because deletion of this exon produces a frame shift in the resulting downstream exons. Thus, CRE can be used to stop Aβ/APP production from the hAβ-KI allele, including in a cell-type specific manner.

To determine if *App* was transcribed at (murine) physiological levels in the hAβ-KI line, RNA was isolated from the brains of homozygous hAβ-KI and WT control mice and analyzed by RNA sequencing (RNA-seq). No difference was detected in *App* expression in brains from young and old WT and hAβ-KI mice (Fig. 1f). Likewise, the level of APP holoprotein was similar from 2 to 22 months of age in hAβ-KI mice (Fig. 1g–i and Supplementary Fig. 8). Furthermore, immunoblot analysis indicated equivalent levels of APP holoprotein in young WT, hAβ-KI (heterozygous) and hAβ-KI (homozygous) mice in multiple brain regions (Fig. 1j–l and Supplementary Fig. 8). As expected, the human-specific antibody 6E10 only detected Aβ in the brains of hAβ-KI mice (Fig. 1j–l and Supplementary Fig. 8). These results are significant because APP is not overexpressed in human sporadic AD, and the hAβ-KI mouse model recapitulates this salient feature.

### Changes in Aβ as a function of age in hAβ-KI mice.
Because amyloid buildup is an integral component of the phenotypic signature of AD, we quantified Aβ levels in mice at different ages. Whereas the amount of soluble Aβ40 and Aβ42 decreased during aging, insoluble Aβ increased, with the highest amount detected between 18 and 22 months of age (Fig. 2a1–a4). However, the Aβ42/40 ratio did not change with age in the hippocampus of the hAβ-KI mice (Supplementary Fig. 2). Thus, the age-related increase in insoluble Aβ could be related to the fact that human and mouse Aβ differ by three amino acids, which accounts for the enhanced amyloidogenicity associated with the human sequence[11]. Aβ aggregation is thought to follow a seeding/nucleation mechanism, involving the slow formation of oligomeric seeds that later grow exponentially[12]. To detect the presence of seeding-competent aggregates in the brain of hAβ-KI mice, we employed the protein misfolding cyclic amplification (Aβ-PMCA) assay[12]. As a positive control, we used brain homogenates from 3xTg-AD mice[13] and as a negative control, brains from WT mice (Fig. 2b, c). Addition of 0.001% of hAβ-KI brain homogenate significantly accelerated Aβ aggregation in this assay, versus the reaction in the presence of the same amount of WT brain homogenate (Fig. 2b). The acceleration in aggregation promoted by hAβ-KI, was similar to that of the positive control consisting of brain homogenate from 3xTg-AD mice, although the lag phase was longer for hAβ-KI-driven aggregation compared to that mediated by 3xTg-AD (Fig. 2b). The $T_{50}$ value, which corresponds to the time needed to reach 50% of maximum aggregation, was shorter for the aggregations performed in the presence of hAβ-KI and 3xTg-AD brain homogenate versus WT mice, indicating a faster aggregation (Fig. 2c) and no difference was observed between hAβ-KI mice (homozygous) and 3xTg-AD mice (homozygous) (Fig. 2c). Despite the increase in insoluble Aβ (including Aβ40 and Aβ42) with age in the hAβ-KI mice and biochemical evidence showing presence of competent aggregative seeds, immunostaining against either Aβ40 or Aβ42 was unable to reliably detect extracellular deposits (Fig. 2d4–d6; 22-month-old mice shown), whereas deposits were readily observed in

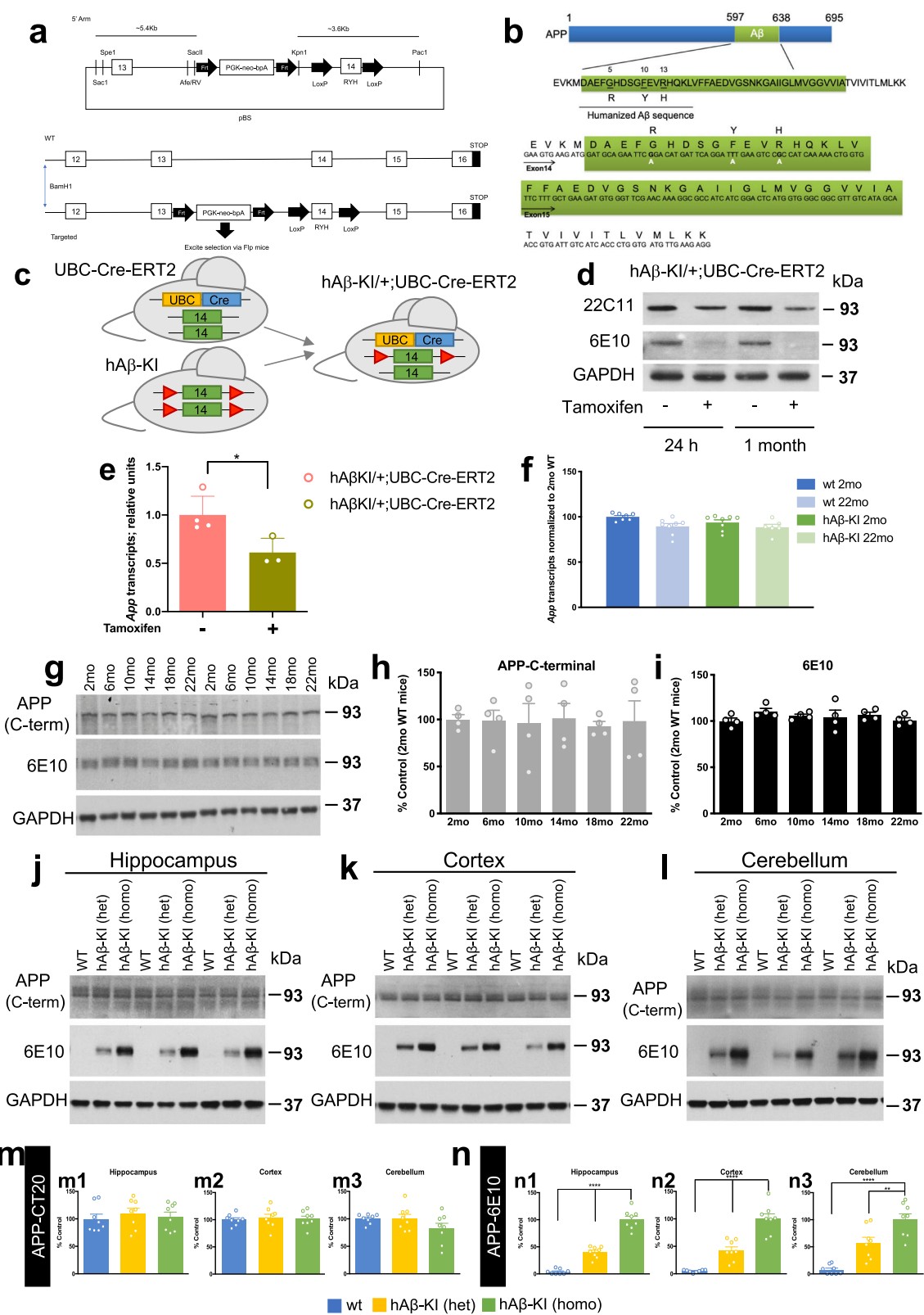

comparable 3xTg-AD brains (Fig. 2d7–d9). As expected, no deposits were observed in WT mice (Fig. 2d1–d3). Likewise, no evidence of thioflavin-S or Congo red-positive deposits was observed in hAβ-KI mice (Fig. 2e, f). In summary, although the brains of hAβ-KI mice displayed an age-associated increase in Aβ and contain competent amyloid seeds, additional factor(s) are required for the formation of amyloid plaques, which suggests

that this model may be useful for introducing risk or other genetic factors to trigger their accumulation.

**OC+ cluster granules are associated with the hippocampal fissure in hAβ-KI mice.** Amyloid protofibrils can be visualized using the conformation specific antibody OC[14]. Immunostaining

**Fig. 1 Humanized Aβ sequence design and APP levels in the hAβ-KI mice. a** Strategy for production of hAβ-KI mice. **b** Schematic representation of humanized Aβ in hAβ-KI mice. **c** Breeding strategy to produce $App^{hA\beta-KI/+}$; UBC-Cre-ERT2 hemizygous mice (green box = exon 14, red triangle = loxP sequence, yellow and blue box = UBC-Cre-ERT2). **d** Western-blot analysis showing that Tamoxifen treatment reduces APP expression (22C11 antibody) in brains of $App^{hA\beta-KI/+}$; UBC-Cre-ERT2 hemizygous mice compared to PBS-treated animals. The hAβ specific product (6E10 antibody) expressed by the floxed hAβ-KI allele is depleted following Tamoxifen treatment. **e** qPCR analysis of *App* expression shows decrease in Tamoxifen-treated $App^{hA\beta-KI/+}$; UBC-Cre-ERT2 hemizygous mice (green) versus PBS-treated mice (pink) (PBS $n = 4$ and tamoxifen $n = 3$; unpaired, two-tailed *t*-test, *$p = 0.035$). **f** No difference in *App* expression in CNS of WT and hAβ-KI homozygous mice at 2 and 22 month of age (transcript level normalized to 2 mo-WT animals; $n = 6$ in hAβ-KI 22 mo, $n = 7$ in WT 2 mo and $n = 8$ in WT 22 mo and hAβ-KI 2 mo) (blue = WT 2 mo, light-blue = WT 22 mo, green = hAβ-KI 2 mo and light-green = hAβ-KI 22 mo). **g–i** Immunoblot analysis of APP recognized by C-terminal APP (CT-20 antibody) (dark-gray) and 6E10 (hAβ-specific) (black) antibody in hippocampal homogenates of hAβ-KI from 2 to 22 months of age ($n = 4$/genotype/age) shows no differences in APP expression. Representative immunoblot analysis of APP holoprotein using APP-CT20 (detect both mouse and human APP) and 6E10 (recognize only human Aβ) in hippocampal (**j**), cortical (**k**) and cerebellar (**l**) homogenates of WT (blue), hAβ-KI (het) (yellow) and hAβ-KI (homo) (green) ($n = 8$/genotype). **m** Quantification of **j**, **k** and **l** shows no difference in APP steady-state level between groups and brain areas. **n** Quantification of **j**, **k** and **l** by 6E10 antibody shows dose-dependent expression of APP in hAβ-KI homozygous mice in the hippocampus (One-way ANOVA, $F_{2,21} = 124.0$, Tukey's post hoc test, ****$p < 0.0001$), cortex (One-way ANOVA, $F_{2,21} = 53.43$, Tukey's post hoc test, ****$p < 0.0001$) and cerebellum (One-way ANOVA, $F_{2,21} = 26.83$, Tukey's post hoc test, **$p < 0.01$ and ****$p < 0.0001$) compared to WT mice and hAβ-KI heterozygous mice. Data are presented as mean values ± SEM.

with OC in brain sections from 22 month-old homozygous hAβ-KI and WT mice revealed the presence of OC+ puncta within the hippocampus of hAβ-KI mice (Supplementary Fig. 3a). OC+ puncta is also observed in 12 month-old 3xTg-AD mice (Supplementary Fig. 3b). These puncta were restricted to the hippocampus, tended to line the hippocampal fissure, and did not represent traditional amyloid-plaques. Confocal microscopy revealed that they were composed of clusters of small granules (Supplementary Fig. 3a). Co-staining for microglia (using antibody IBA1) showed that these clusters of granules were not associated with microglia, and do not appear to induce any changes in microglial phenotypes in their immediate vicinity (Supplementary Fig. 3a). However, cellular processes were observed surrounding the clusters (highlighted by yellow arrows), which also did not colocalize with microglia. Immunostaining with OC and GFAP, an astrocyte marker, showed a clear association between the OC+ granules and the processes of a single astrocyte (Fig. 3a). Notably, OC+ granules do not appear to reside intracellularly within an astrocyte but are most likely associated with astrocytic processes and projections (Fig. 3a4b–a6b). The number of OC+ granule clusters increased with age in both WT and hAβ-KI mice (Fig. 3a, b). However, there were fewer OC+ clusters in WT mice, and these were less prominent and dimmer than those seen in hAβ-KI hippocampi (Fig. 3a, b). Clusters were present at 10 months of age in hAβ-KI mice, and persisted thereafter (Fig. 3a; clusters highlighted by white arrows; Fig. 3b). Clusters of granules that associate with astrocytes specifically in the hippocampus have been described previously in both aged mice and accelerated brain aging mouse models[15,16] and have been identified using a Periodic Acid Schiff (PAS) stain for polysaccharides. Accordingly, we performed a PAS stain in WT and hAβ-KI mouse brain sections and were able to show the same distinct clusters of granules (Supplementary Fig. 3b). Staining for OC+ granules across the rostral-caudal axis of the hAβ-KI brain revealed the presence of clusters in locations other than in the hippocampus, notably in adjacent outer cortical regions starting at the anterior olfactory nucleus and extending caudally through the olfactory tubercle, piriform cortex, through to the cochlear nucleus (Fig. 3c). Confocal imaging of the piriform cortex showed the same clusters of granules, with each cluster associated with astrocytic processes (Fig. 3d), with affected astrocytes lining the cortical plate and the granules seemingly projected towards the cortical surface.

To determine if the presence of OC+ clusters in hAβ-KI mice was correlated with expression of human Aβ, 2-month-old hAβ-KI homozygous; UBC-CRE^ERT2 hemizygous mice were treated with tamoxifen or sunflower oil (vehicle) to induce CRE-

dependent recombination of the *loxP* sites to disrupt Aβ/APP production (Fig. 3e). Indeed, at 8 months of age, tamoxifen-treated hAβ-KI homozygous; UBC-CRE^ERT2 hemizygous mice showed fewer OC+ clusters versus PBS-treated animals of the same genotype (Fig. 3f, g). These results indicate that expression of human Aβ in hAβ-KI mice increases the number of these structures compared to WT mice.

**Presenilin mutation accelerates OC+ granules formation in the hAβ-KI mice.** Because hAβ-KI mice accumulate OC+/PAS+ clusters of granules in the hippocampus, we investigated whether co-expression of familial-linked AD mutations affected the formation of OC+ granules in hAβ-KI homozygous mice by breeding hAβ-KI and PS1^M146V mice to homozygosity and aging them to 18 months (Fig. 4a). Quantification of soluble and insoluble Aβ showed increased Aβ (Aβ40 and Aβ 42) in double-homozygous hAβ-KI/PS1^M146V compared with homozygous hAβ-KI mice, (Fig. 4b). Examination of OC+ granules in the hippocampus of WT, homozygous hAβ-KI, and double-homozygous hAβ-KI/PS1^M146V mice (Fig. 4c1–c3) showed increased OC+ clusters in hAβ-KI/PS1^M146V mice compared to the other groups, with each cluster of granules being in close association with astrocytes (Fig. 4c1b–c3b). Quantification of the number and characteristics of the clusters across the three genotypes was performed, revealing a stepwise increase in the number of clusters per hippocampus (WT -> hAβ-KI -> hAβ-KI/PS1^M146V; Fig. 4d), the average number of granules within a cluster (Fig. 4e), the average area of an individual granule (Fig. 4f), and in the average area encompassed by a single cluster (Fig. 4g). No evidence of fibrillar amyloid aggregates was observed in the homozygous hAβ-KI/PS1^M146V mice and in association with the granules (Fig. 4c, 4cb). Overall, these results suggest that an increase in Aβ levels induced by the presenilin mutation M146V accelerates the formation of OC+ clusters in double homozygous hAβ-KI/PS1^M146V mice.

**Cognitive, synaptic and inflammatory alterations are present in hAβ-KI mice.** We next assessed the impact of human Aβ expression on memory-related processes. Homozygous hAβ-KI mice were impaired in cortical- and hippocampal-dependent cognitive performance versus WT controls, indicating that age-dependent buildup of human Aβ expressed from the humanized *App* locus is sufficient to impair cognition in mice (Fig. 5a, b). Notably, impairments were detected from 10 to 14 months of age, corresponding to the appearance of the OC+/PAS granules seen in the hAβ-KI brains. To determine whether key aspects of

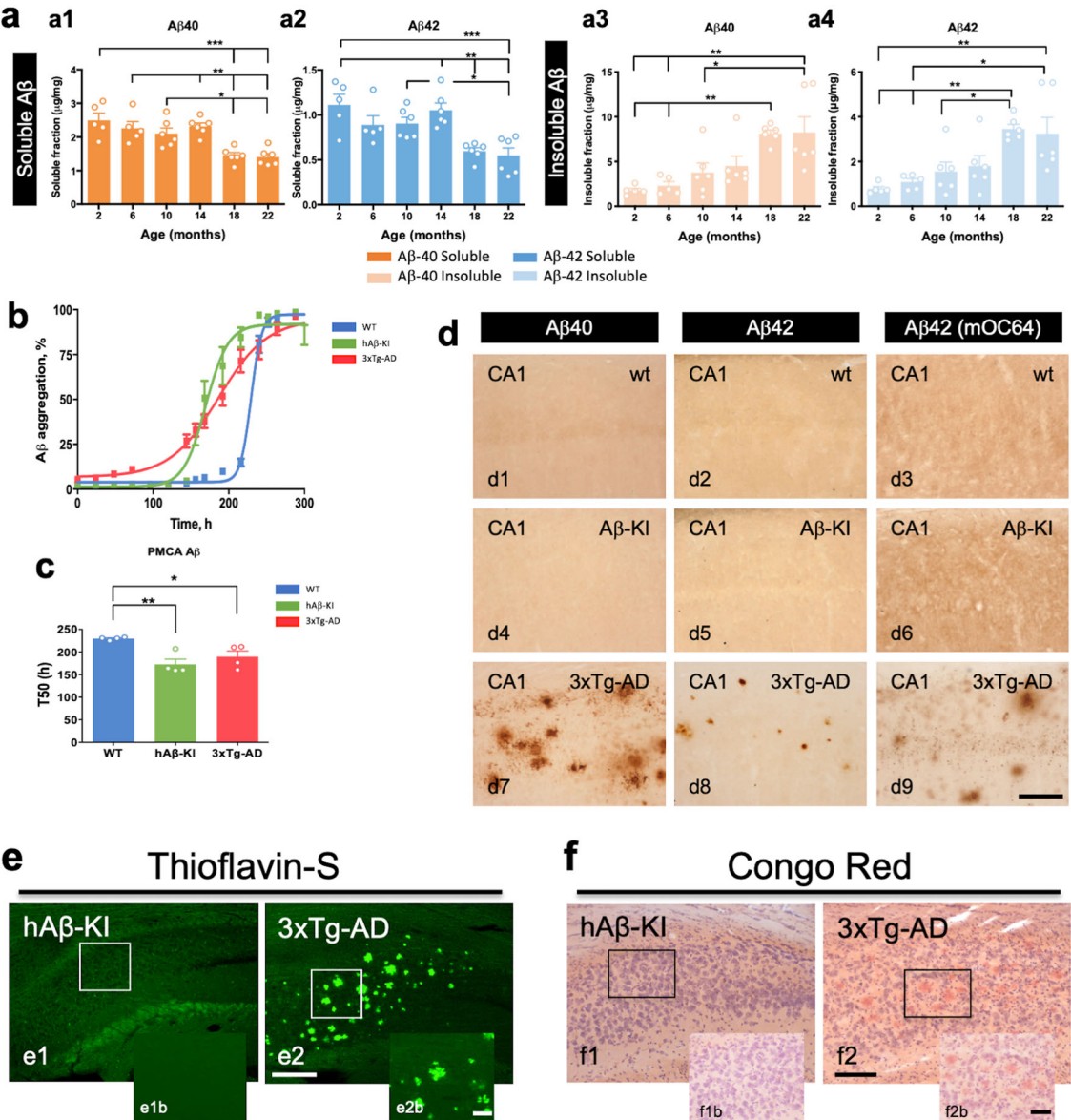

**Fig. 2 Changes in Aβ as a function of age in hAβ-KI mice. a** Aβ40 (orange) and Aβ42 (blue) quantity in hAβ-KI homozygous mice was determined using the MSD V-PLEX Plus Aβ Peptide Panel 1 (6E10) Kit ($n = 5$ in 2 mo and 6 mo and $n = 6$ in 10 mo, 14 mo, 18 mo and 22 mo). The ELISA analysis shows that hAβ-KI mice produce high soluble Aβ levels, including Aβ40 (One-way ANOVA, $F_{5,28} = 10.01$, Tukey's post hoc test, *$p < 0.05$, **$p < 0.01$ and ***$p < 0.001$) and Aβ42 (One-way ANOVA, $F_{5,28} = 8.057$, Tukey's post hoc test, *$p < 0.05$, **$p < 0.01$ and ***$p < 0.001$) compared to old hAβ-KI mice (a1 and a2). Opposite effect is observed in insoluble Aβ levels in older hAβ-KI mice compared to younger hAβ-KI mice, including in Aβ40 (One-way ANOVA, $F_{5,28} = 7.21$, Tukey's post hoc test, *$p < 0.05$ and **$p < 0.01$) and Aβ42 (One-way ANOVA, $F_{5,28} = 6.543$, Tukey's post hoc test, *$p < 0.05$ and **$p < 0.01$) (a3-a4). **b** Aβ-PMCA shows accelerated in vitro aggregation using hAβ-KI (green) and 3xTg-AD (red) brain homogenate to seed monomeric Aβ compared to WT (blue) controls ($n = 4$/genotype). **c** Shorter time to reach 50% of total aggregation, measured by thioflavin-T emission levels ($n = 4$/genotype) (One-way ANOVA, $F_{2,11} = 9.083$; Tukey's post hoc test, *$p < 0.05$ and **$p < 0.01$). **d** Immunohistochemistry performed with Aβ40 and Aβ42 antibodies showed no sign of aggregates (plaques) for Aβ isoforms in hAβ-KI mice at 22-month of age (d4-d6). As a positive control, 3xTg-AD mice were used with significant Aβ40 and Aβ42 aggregates present at 22-month of age (d7-d9). WT mice showed no staining for both Aβ isoforms (d1-d3). 22-month-old hAβ-KI mice and 3xTg-AD mice were stained with thioflavin-S (Thio-S) (**e**) or Congo Red (**f**) stain. No fibrillar aggregated stained with Thio-S or Congo red is observed in hAβ-KI (e1 and f1). As a positive control, 3xTg-AD mice were used with fibrillary extracellular aggregates positive for Thio-S and Congo Red present in the hippocampus (e2 and f2). Data are presented as mean values ± SEM. Scale bar: 200 μm (e1, e2, f1 and f2), 100 μm (d1-d9) and 50 μm (e1b, e2b, f1b and f2b).

synaptic physiology are impacted in homozygous hAβ-KI mice, we examined the effects on long-term potentiation (LTP), a form of synaptic plasticity, as well as on baseline synaptic transmission. Theta-burst stimulation-induced LTP revealed deficits in the field excitatory postsynaptic potentials (fEPSPs) 50–60 min post-induction in slices from male and female homozygous hAβ-KI mice versus slices from WT mice at 18 months of age (Fig. 5c).

Thus, it appears that the LTP impairment involves events that occur after induction and initial expression of potentiation, a conclusion consistent with the observation that short-term potentiation (i.e., the first 2 min following induction) was unaffected. No LTP impairment was observed in 2-month-old mice (Fig. 5c). Moreover, quantification of synaptic puncta density showed a significant reduction at presynaptic level (recognized

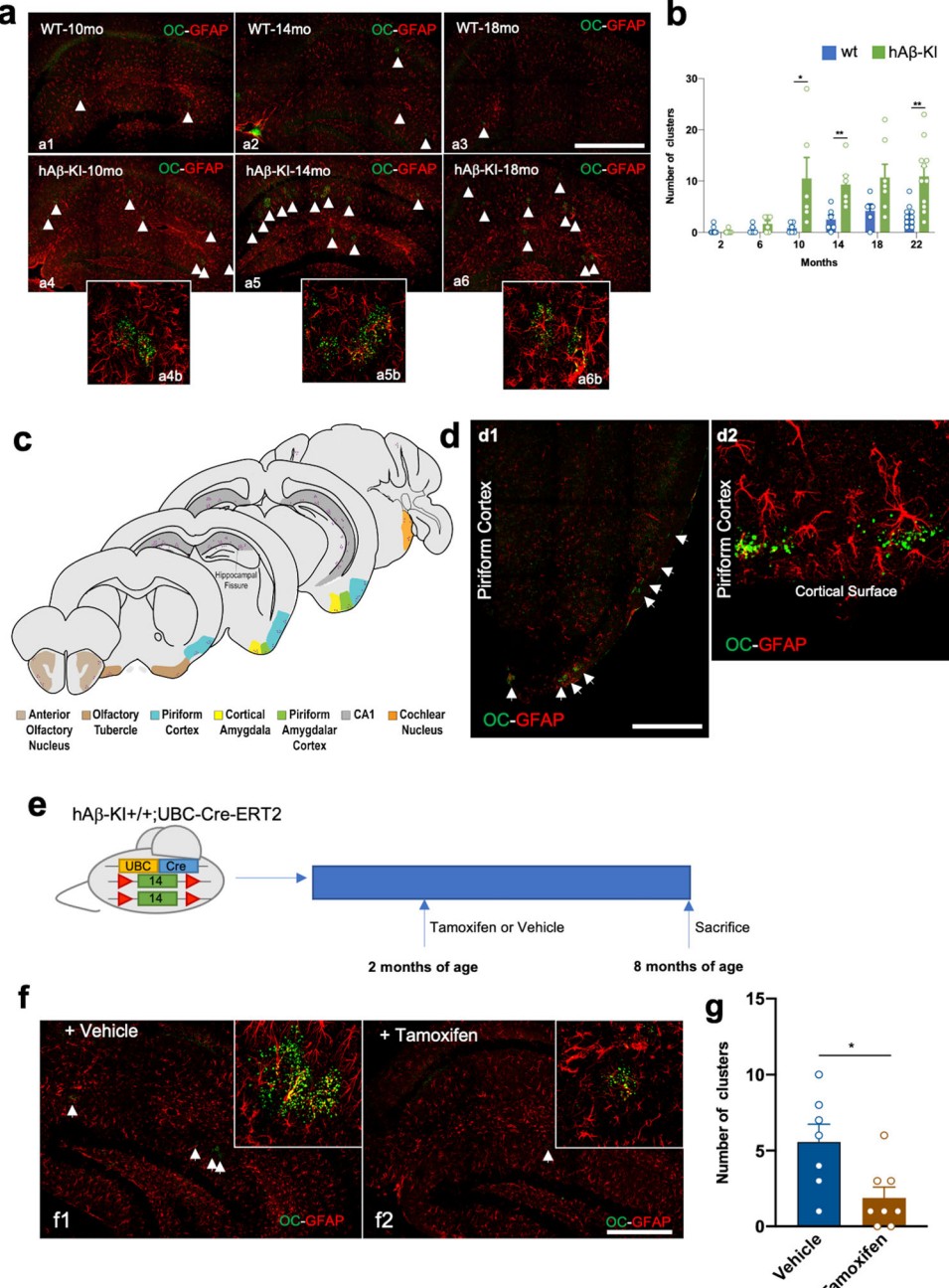

**Fig. 3 Human wild-type Aβ exacerbates the formation of OC+ clusters in the hippocampus of hAβ-KI mice. a** Representative images for astrocytes and OC+ cluster in WT (a1–a3) and homozygous hAβ-KI (a4–a6) mice across their lifespans (individual clusters highlighted by white arrowheads). Confocal images showed the association of the astrocytes with OC+ granules (a4b–a6b) in hAβ-KI mice. **b** Quantification of the number of OC+ clusters/ hippocampal section at each age in homozygous hAβ-KI (green) and WT (blue) mice (WT 2 mo = 16, 6 mo = 6; 10 mo = 6; 15 mo = 6; 18 mo = 5; 22 mo = 10; hAβ-KI 2 mo = 9; 6 mo = 6; 10 mo = 6, unpaired, two-tailed t-test, *p = 0.0411; 14 mo = 6, unpaired, two-tailed t-test, **p = 0.0070; 18 mo = 7; 22 mo = 11, unpaired, two-tailed t-test, **p = 0.0022). **c** Schematic representation of OC+ clusters of granules distribution in the brain of hAβ-KI mice (light brown = Anterior Olfactory Nucleus, brown = Olfactory Tubercle, blue = Piriform Cortex, yellow = Cortical Amygdala, green = Piriform Amygdalar Cortex, gray = CA1 and orange = Cochlear nucleus). Graphic adapted from Image 26, 47, 73, 81 and 110 of the Allen Mouse Brain Atlas. Copyright 2004, Allen Institute for Brain Science. Available from https://mouse.brain-map.org/static/atlas[63]. **d** Immunostaining for astrocytes (GFAP, red channel) and protofibrils (OC, green channel) shows their accumulation in the piriform cortex, lining the cortical edge in hAβ-KI (d1) and WT (d2) mice. **e** Experimental design in which hAβ-KI homozygous; UBC-CRE[ERT2] hemizygous mice are treated with tamoxifen or vehicle for 6 months (green box = exon 14, red triangle = loxP sequence, yellow and blue box = UBC-Cre-ERT2). Mice were sacrificed at 8-month of age. **f** Immunostaining for astrocytes (GFAP, red) and protofibrils (OC, green) in the hippocampus of either vehicle or tamoxifen-treated hAβ-KI homozygous; UBC-CRE[ERT2] hemizygous mice. **g** Quantification of OC+ granular clusters in vehicle (blue) and Tamoxifen (brown) treated hAβ-KI homozygous; UBC-CRE[ERT2] hemizygous mice (Vehicle n = 7, Tamoxifen n = 8; unpaired, two-tailed t-test, *p = 0.0160). Data are presented as mean values ± SEM. Scale bar: 300 μm (a1–a6 and f1–f2) and 200 μm (d1–d2).

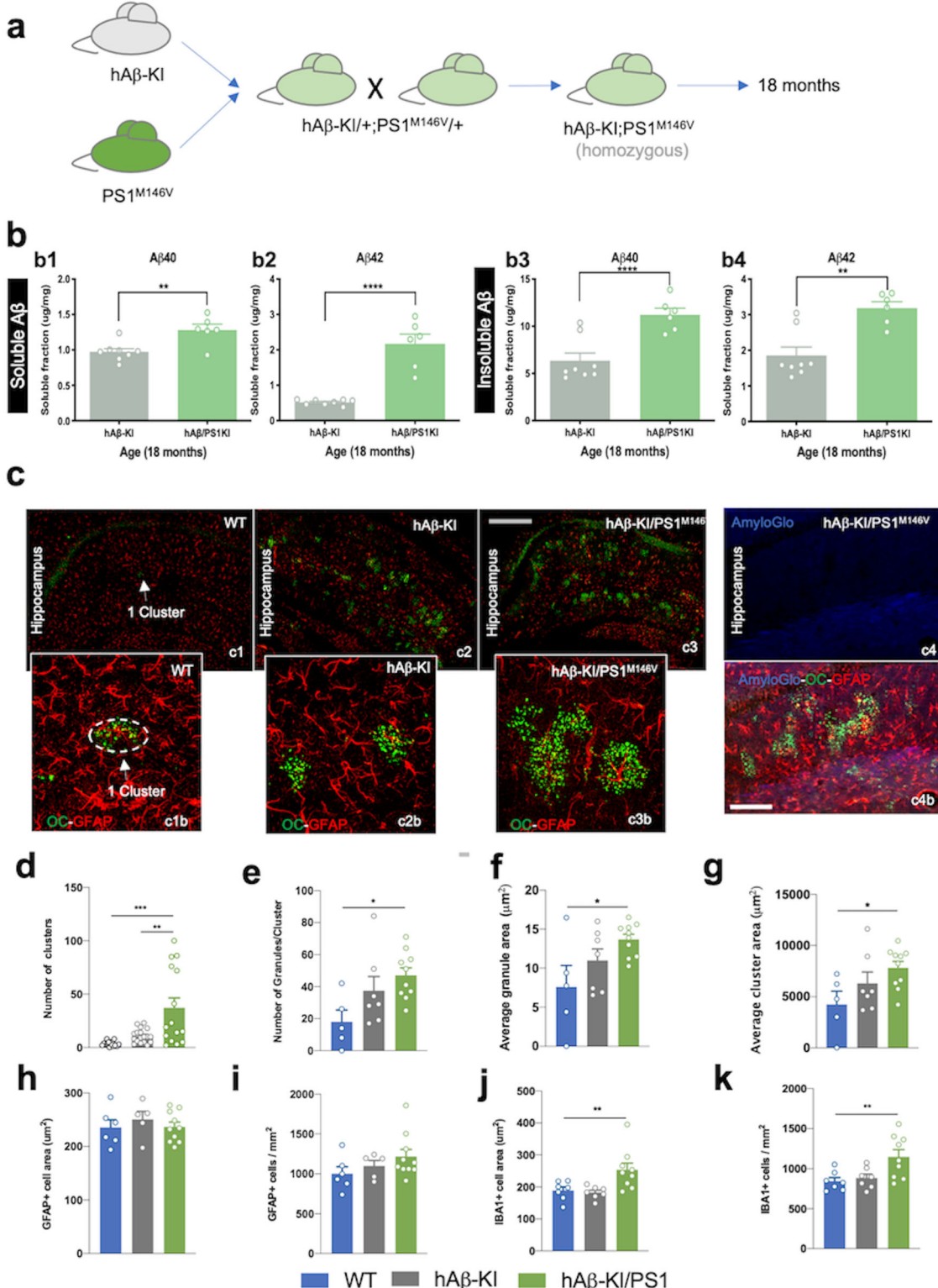

with the antibody synaptophysin) in the hAβ-KI compared to WT mice at 18 months of age (Fig. 5d, e). Despite cognitive and synaptic loss is observed in this line, no cell loss was detected in the pyramidal layer of the CA1 hippocampal region in the hAβ-KI compared to WT mice at 22 months of age (Supplementary Fig. 4a, b). However, a significant decrease in the hippocampal volume was observed in the hAβ-KI line at 22 months of age compared to WT mice (Fig. 5f). Overall, these data show that substitution of mouse Aβ with wildtype human Aβ induces age-

dependent changes in both cognition and synaptic plasticity in mice even when expressed at murine physiological levels.

In designing the hAβ-KI mice, we floxed the Aβ-encoding exon 14 to permit Cre-mediated ablation of Aβ expression. To determine if a higher-level neurobiological function such as cognition could be rescued, we bilaterally infused adeno-associated virus containing either a control or CAMKII-Cre vector construct into the hippocampus of hAβ-KI mice (Fig. 6a, b). After one month, homozygous hAβ-KI mice treated with

**Fig. 4 OC+ clusters formation are exacerbated by presenilin 1 mutation in hAβ-KI mice. a** hAβ-KI mice (gray) are bred to homozygosity with PS1$^{M146V}$ mice (green) and aged to 18 months. **b** Aβ40 and Aβ42 levels in hAβ-KI (dark-gray) and hAβ-KI/PS1$^{M146V}$KI (green) homozygous mice were quantified using the MSD V-PLEX Plus Aβ Peptide Panel 1 (6E10) Kit ($n = 6$ in hAβ-KI/PS1$^{M146V}$KI group and $n = 8$ in hAβ-KI group). Soluble Aβ40 (b1, unpaired, two-tailed $t$-test, **$p = 0.0041$) and Aβ42 (b2, unpaired, two-tailed $t$-test, ****$p < 0.0001$) and insoluble Aβ40 (b3, unpaired, two-tailed, $t$-test ***$p = 0.0009$) and Aβ42 (b4, unpaired, two-tailed $t$-test, **$p = 0.0013$) showed an increase in hAβ-KI/PS1$^{M146V}$KI compared to hAβ-KI mice. **c** Immunostaining for astrocytes (GFAP, red) and protofibrils (OC, green) shows accumulation of OC+ clusters in the hippocampus (high magnification images; c1b–c3b). AmyloGlo stain (blue channel) in hAβ-KI/PS1$^{M146V}$KI homozygous mice (c4) shows that this line did not generate fibrillar aggregates. **d–g** Quantification of (**c**) showing the number of clusters per hippocampal slice (**d**) the average number of OC+ granules within an individual cluster (WT $n = 6$, hAβ-KI $n = 18$, hAβ-KI/PS1$^{M146V}$KI $n = 15$; ANOVA, $F_{2,45} = 13.11$, Tukey's post hoc test, **$p < 0.01$ and ***$p < 0.001$) (**e**) the average area of a single granule (WT $n = 5$, hAβ-KI $n = 7$, hAβ-KI/PS1$^{M146V}$KI $n = 10$, One-way ANOVA, $F_{2,19} = 0.1043$, Tukey's post hoc test, *$p < 0.05$), (**f**) and the average area taken up by a cluster (WT $n = 5$, hAβ-KI $n = 7$, hAβ-KI/PS1$^{M146V}$KI $n = 10$, One-way ANOVA, $F_{2,19} = 2.534$, Tukey's post hoc test, *$p < 0.05$), (**g**) in 18-month WT (blue), hAβ-KI (dark-gray) and hAβ-KI/PS1$^{M146V}$KI (green) mice (WT $n = 5$, hAβ-KI $n = 7$, hAβ-KI/PS1$^{M146V}$KI $n = 10$, One-way ANOVA, $F_{2,19} = 0.1639$, Tukey's post hoc test, **$p < 0.01$). **h–k** Densitometric analysis using Imaris software to quantify density (**h**) and cell area (WT $n = 6$, hAβ-KI $n = 7$, hAβ-KI/ PS1$^{M146V}$KI $n = 9$) (**i**) of GFAP+ astrocytes or the density (**j**) and cell area (**k**) of IBA1+ microglia cells in 18-month WT (blue), hAβ-KI (dark-gray) and hAβ-KI/PS1$^{M146V}$KI (green) mice (WT $n = 7$, hAβ-KI $n = 7$, hAβ-KI/PS1$^{M146V}$KI $n = 9$, One-way ANOVA, $F_{2,20} = 1.716$, Tukey's post hoc test, **$p < 0.01$ and $F_{2,20} = 4.435$, Tukey's post hoc test, **$p < 0.01$). Data are presented as mean values ± SEM. Scale bar: 300 μm (c1–c3) and 100 μm (c4–c4b).

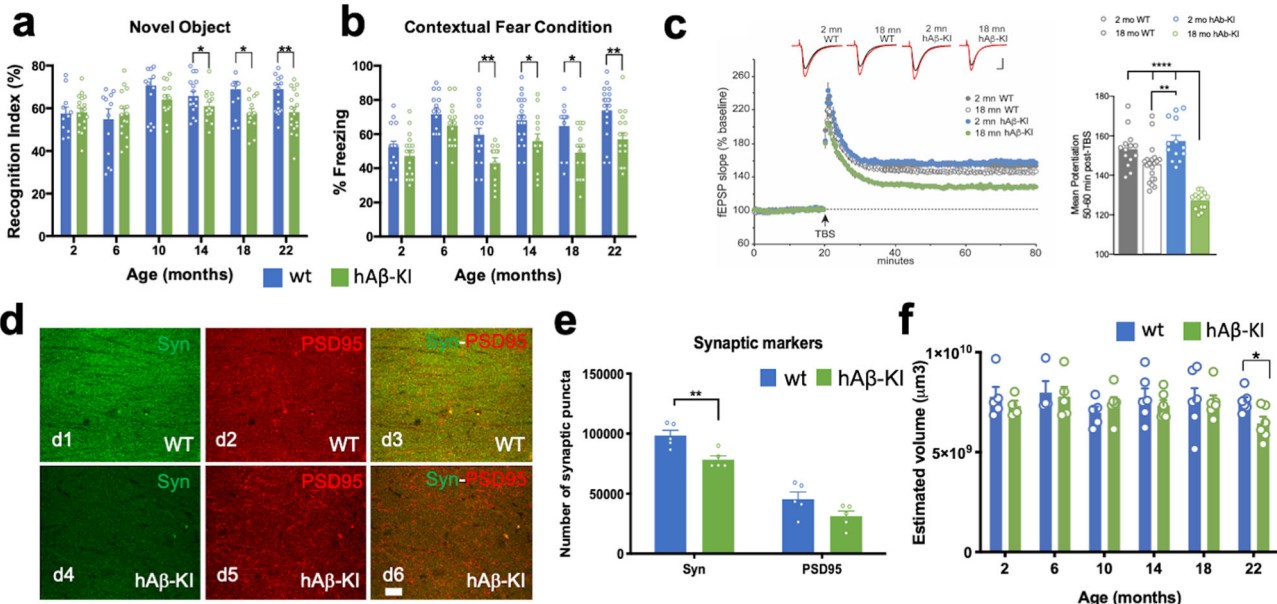

**Fig. 5 Phenotypic alterations in CNS of hAβ-KI mice. a** hAβ-KI (green) mice showed cortical deficits at 14, 18 and 22 months compared to WT (blue mice) (unpaired, two-tailed $t$-test, 14 mo *$p = 0.0381$, 18 mo *$p = 0.0131$ and 22 mo **$p = 0.0024$) (WT 2 mo $n = 11$, 6 mo $n = 13$, 10 mo and 14 mo $n = 16$, 18 mo $n = 12$, and 22 mo $n = 20$; hAβ-KI 2 mo, 18 mo and 22 mo $n = 18$, 6 mo $n = 16$, 10 mo $n = 13$ and 14 mo $n = 14$). **b** Hippocampal deficits were also observed in hAβ-KI (green) mice at 10, 14, 18 and 22 months compared to WT (blue) mice (unpaired, two-tailed $t$-test, 10 mo **$p = 0.0037$, 14 mo *$p = 0.0239$, 18 mo *$p = 0.0112$ and 22 mo **$p = 0.0020$) (WT 2 mo $n = 15$, 6 mo $n = 14$, 10 mo $n = 18$, 14 mo $n = 20$, 18 mo $n = 12$, 22 mo $n = 19$; hAβ-KI 2 mo and 22 mo $n = 18$, 6 mo and 18 mo $n = 15$, 10 mo $n = 13$ and 14 mo $n = 14$). **c** Acute hippocampal slices were examined for changes in synaptic plasticity in 2 and 18-month-old WT (gray = 2 mo and white = 18 mo) and hAβ-KI (blue = 2 mo and green = 18 mo) mice. Theta burst induced LTP was impaired in slices from hAβ-KI mice (2 mo and 18 mo $n = 13$ slices) relative to aged-matched WT controls (2 mo $n = 14$ slices, 18 mo $n = 20$ slices). Synaptic responses in control pathway remained stable (no theta stimulation) throughout the recording session. Insets show field synaptic responses collected during baseline (black line) and 1 h after theta burst stimulation (red line). Scale: 1 mV/5 ms. Mean (±SEM) percent potentiation 50–60 min post-TBS was markedly depressed in slices from hAβ-KI mice relative to WT controls (two-way ANOVA, major effect in interaction $F_{1,56} = 22.46$ and age factor $F_{1,56} = 64.52$, Tukey's post hoc test, **$p < 0.01$ and ****$p < 0.0001$). **d, e** Confocal images of synaptic puncta stained with synaptophysin (green) and PSD-95 (red) antibodies, shows a significant decrease at presynaptic level in hAβ-KI (d4–d6; green) versus WT (d1–d3; blue) mice at 18 months (unpaired, two-tailed $t$-test, **$p = 0.0062$) ($n = 5$/genotype). **f** Cavalieri method shows significant differences in hippocampal volume between WT (blue) and hAβ-KI (green) mice at 22 months of age (unpaired, two-tailed $t$-test, *$p = 0.0199$; $n = 4$ in 6 mo-WT and 2 mo-hAβ-KI, $n = 5$ in 2 mo-WT, 10 mo-WT and 6 mo-hAβ-KI, $n = 6$ in 14 mo, 18 mo, 22 mo-WT and 10 mo, 14 mo, 18 mo and 22 mo-hAβ-KI). Data are presented as mean values ± SEM. Scale bar: 10 μm (d1–d6).

control vector (hAβ-KI-C.V) continued to show significant impairment in the object location memory test (OLM) versus WT mice infused with the control vector (WT-C.V) (Fig. 6c). Remarkably, homozygous hAβ-KI mice treated with AAV-CAMKII-Cre construct (hAβ-KI-Cre) showed significant recovery in the OLM test versus hAβ-KI mice infused with the control

vector, suggesting that depletion of Aβ reverses the cognitive deficits observed in these mice (Fig. 6c). Soluble Aβ levels did not change in a significant manner between hAβ-KI-C.V mice versus hAβ-KI-Cre mice (Fig. 6d1, d2), whereas insoluble Aβ40 and Aβ42 levels showed a robust reduction in hAβ-KI-Cre mice versus hAβ-KI-C.V mice (Fig. 6d3, d4). These results

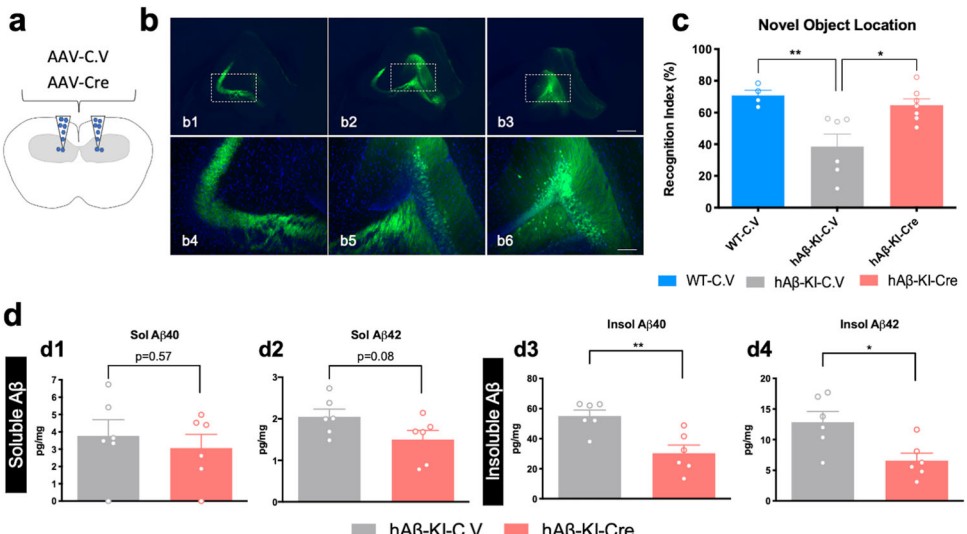

**Fig. 6 Cre-mediated deletion of Aβ encoding exon ameliorates the hippocampal memory deficits in hAβ-KI mice. a** Schematic diagram of AAV delivery in the hippocampus (coronal section). **b** Representative images of AAV-CAMKII-GFP construct expression in the hippocampus. **c** Cognitive performance (OLM) of 25-mo WT and hAβ-KI homozygous mice treated with control AAV-CAMKII and experimental AAV-CAMKII-Cre vectors (One-way ANOVA, $F_{2,14} = 5.971$; Tukey's post hoc test, \*$p < 0.05$ and \*\*$p < 0.01$)) (blue = WT-C.V, gray = hAβ-KI-C.V and red = hAβ-KI-Cre). ($n = 4$ in WT-C.V, $n = 6$ in hAβ-KI-C.V and $n = 7$ in hAβ-KI-Cre). **d** Hippocampal Aβ40 and Aβ42 was quantified by MSD V-PLEX Plus Aβ Peptide Panel 1 (6E10) Kit ($n = 6$/ genotype/treatment) (gray = hAβ-KI-C.V and red = hAβ-KI-Cre), showing a significant decrease in soluble Aβ40 (unpaired, two-tailed t-test, $p = 0.057$) and Aβ42 (unpaired, two-tailed t-test, $p = 0.08$) and insoluble Aβ40 (unpaired, two-tailed, t-test \*\*$p = 0.0042$) and Aβ42 (unpaired, two-tailed t-test, \*$p = 0.0149$) in hAβ-KI-Cre compared to hAβ-KI-C.V mice. Data are presented as mean values ± SEM. Scale bar: 400 µm (b1–b3), 100 µm (b4–b6). AAV-CAMKII-GFP construct were defined in the figures as control vector (C.V) and AAV-CAMKII-Cre as Cre.

demonstrate that the humanized Aβ produces a significant cognitive deficit and that suppression of its expression mitigates these impairments.

Inflammatory processes are an important hallmark associated with AD[17]. We screened for evidence of inflammation in the hippocampus of the hAβ-KI mice. Soluble pro-inflammatory cytokines, such as IL-1β and TNFα, increased in hAβ-KI mice with aging whereas anti-inflammatory cytokines such as IL-2, IL-4 and IL-10 were decreased (Supplementary Fig. 5a). Similar trends were observed in pro-inflammatory cytokines when hAβ-KI mice were compared to WT mice (Supplementary Fig. 5b). However, anti-inflammatory cytokines were reduced in old hAβ-KI mice versus the other groups (Supplementary Fig. 5b). Collectively, these data suggest that alterations of the immune response are also present in this new animal model. The changes in anti-inflammation-related cytokines were not associated with alterations in microglia or astroglia cell density (Supplementary Fig. 6a, b1, c and d1). However, morphological quantifications showed more immune-reactive astrocytes in old (22 mo) hAβ-KI mice versus younger (2 mo) hAβ-KI mice and WT mice (Supplementary Fig. 6a, b2–b4). No differential morphological alteration was observed in microglia (Supplementary Fig. 6c, d2–d4). These results raise the possibility that immune-reactive astrocytes are involved in formation of OC+/PAS granules. In summary, these data support that anti-inflammatory cytokines decrease with aging but that pro-inflammatory cytokines increase[18]. In addition, it appears that astrocytes, and not microglia, are the predominant reactive cell type observed in these mice and associated with PAS granules. These initial changes in the astrocytic population are consistent with a previous study that reactive astrocytes occur in the hippocampus even before the onset of the symptomatic phase of AD[19]. Together, these studies suggest that earlier changes in astrocytic biology occur in our new model as in AD and that this may contribute to the generation of OC+/PAS aggregates.

**Metabolic, energetic and glutamatergic/ion channels modules are altered in hAβ-KI mice and AD patients.** To investigate the consequence of expressing humanized Aβ on overall gene expression, bulk RNA-seq was performed on transcripts extracted from hippocampal tissue. This analysis identified 5 differentially expressed genes between WT and hAβ-KI mice at 2 months of age (Fig. 7a). Age-related changes in gene expression were also observed between younger and older mice, within both the hAβ-KI line and the WT controls (Fig. 7b, c). Most notably, significant differences in gene expression were uniquely present in ~14 genes in older (22-mo-old) hAβ-KI mice relative to age-matched WT controls (Fig. 7d). These genetic differences were also found in the younger hAβ-KI and WT controls (Fig. 7e). The altered genes included those involved in metabolism (*Dhcr7, Sdhd, Hspe1 and Wdfy1*), neuroplasticity/neurotransmission (*Atp2b1, Gabra2, Lppr4, Tppp3, Diras2 and Nsmf*), as well as changes in transcriptional and splicing-related factors (*Chd4, Psip1 and R3hdm4*) (Fig. 7e). These findings indicate that expression of humanized Aβ at physiological levels is sufficient to trigger alterations in expression of genes involved in key biological processes, including metabolism, synaptic plasticity and memory-related functioning.

Furthermore, we applied weighted gene co-expression network analysis (WGCNA)[20] to place transcriptomic changes in hAβ-KI mice within a system-level framework. WGCNA has been successfully used to unravel meaningful biological relationships associated with disease in human and mouse model systems[21–23]. WGCNA identifies modules consisting of highly correlated genes, in which the module eigengene, or the first principal component of module gene expression, can be used as a module summary. To identify which modules provide the most biological explanatory power, modules are annotated as to whether their expression is correlated to disease status, whether genes specific to these modules are enriched in specific biological pathways or cells types, and whether there are potential confounders[22,24,25]. We identified three mRNA modules significantly associated with

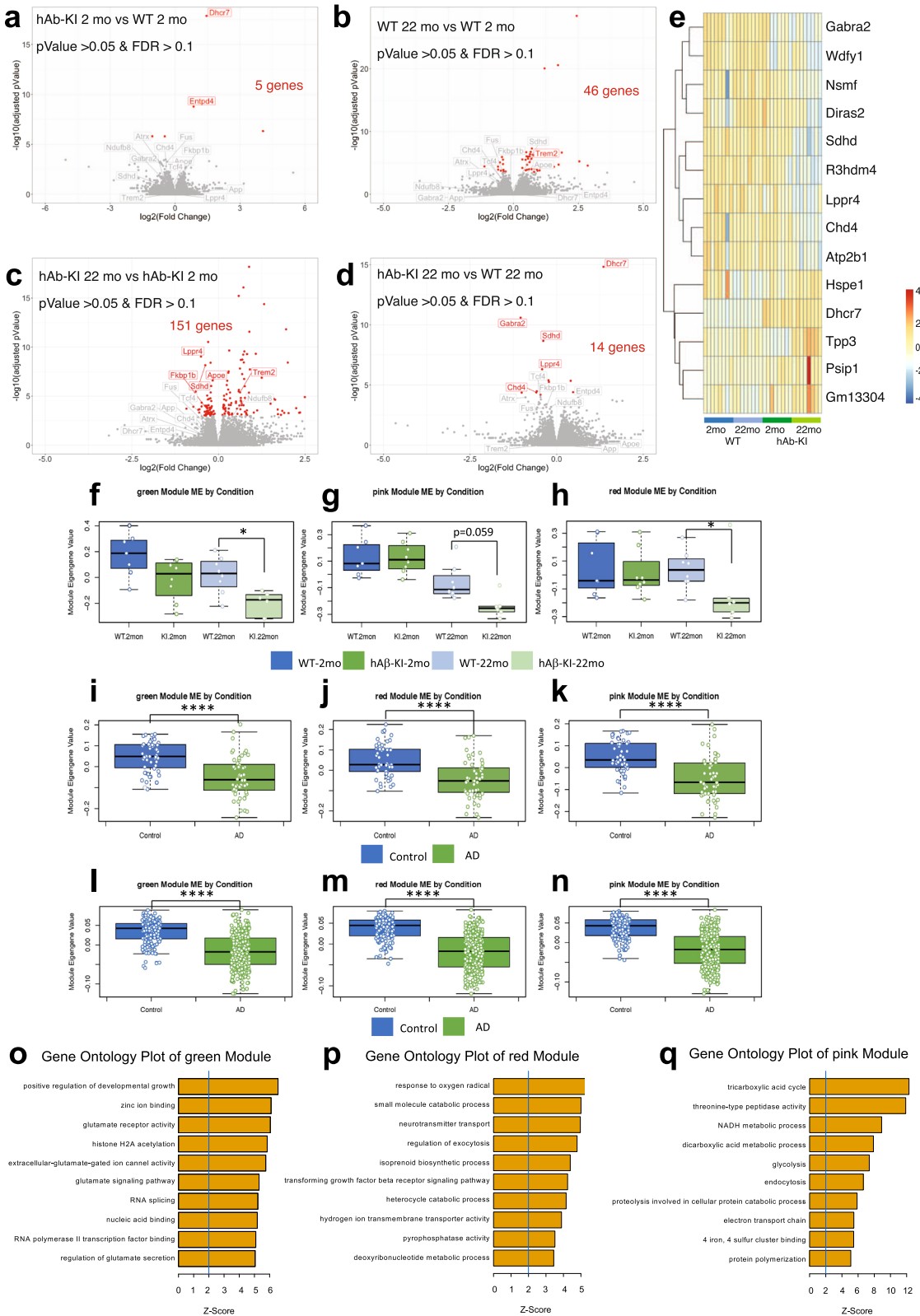

homozygosity of hAβ-KI – labeled green, red and pink (FDR $q <$ 0.05, Supplementary Fig. 7). These modules are downregulated in hAβ-KI mice at 22-months only but not at 2-months (Fig. 7f–h), consistent with histopathological changes in the hAβ-KI mice at 22-months. Cell-type enrichment analysis demonstrated that the red and pink modules are enriched in neuronal marker genes, but the green module is not enriched in any cell-type specific markers

(Supplementary Fig. 7b). Functional annotation of the green module shows that it is enriched in gene-ontology (GO) terms related to glutamate-ion channel binding and signaling and histone acetylation (Fig. 7o and Supplementary Fig. 7c). One of the hub genes of the green module is *Gria2*, a GluR2 subunit of the AMPA receptor, which is involved in cell migration and calcium signaling[26]. The green module also includes other genes

**Fig. 7 Transcriptomic analysis in hAβ-KI mice.** Differential gene expression analysis was performed comparing 2 months WT and hAβ-KI mice (**a**), 2 and 22 months WT mice (**b**), 2 and 22 months hAβ-KI mice (**c**) and 22 months WT and hAβ-KI mice (**d**), differentially expressed gene were selected following $p < 0.05$ and FDR < 0.1 values ($n = 6$ in hAβ-KI 22 mo, $n = 7$ in WT 2 mo and $n = 8$ in WT 22 mo and hAβ-KI 2 mo). **e** Heatmap of group d of genes with $p < 0.05$ and FDR < 0.1 were represented with expression data from the 2 months WT and hAβ-KI mice added. Module eigengene trajectory of green (**f**), pink (**g**) and red (**h**) modules in WT and hAβ-KI mice at 2 and 22 months ($n = 6$ in hAβ-KI 22 mo, $n = 7$ in WT 2 mo and $n = 8$ in WT 22 mo and hAβ-KI 2 mo) (blue = WT 2 mo, light-blue = WT 22 mo, green = hAβ-KI 2 mo and light-green = hAβ-KI 22 mo). Module eigengene trajectory of green (**i**), red (**j**) and pink (**k**) modules in human temporal cortex AD data ([28], $n = 160$ samples) (blue = control and green = AD). Module eigengene trajectory of green (**l**), red (**m**) and pink (**n**) modules in human AD frontal cortex data ([64], $n = 680$ samples) (blue = control and green = AD). In all the boxplots, the upper and lower lines represent the 75th and 25th percentiles, respectively, while the center line represents the median (unpaired, two-tailed, t-test *$p < 0.05$ and ****$p < 0.0001$ were used). Gene ontology term enrichment for green (**o**), red (**p**) and pink (**q**) modules.

involved in glutamate signaling including *Grin2a*, *Gria3* and *Grm1*. The red module, which is also downregulated in hAβ-KI mice at 22-months, is enriched in GO terms related to neurotransmitter transport and metabolic processes (Fig. 7p and Supplementary Fig. 7d). The downregulated pink module is enriched in mitochondrial energetics-associated GO terms and include genes like *Aldoa*, an aldolase and *Cx3cl1* (Fractalkine) (Fig. 7q and Supplementary Fig. 7e).

To investigate the possible relevance of these changes in gene expression to those occurring in human AD, we determined whether the hAβ-KI associated modules are preserved in human AD cases. Using module preservation analysis, we found that green, red and pink modules are highly represented in human AD data ($Z_{summary} > 5$, Supplementary Fig. 7f)[22,27]. Using synthetic module-eigengene revealed that green, red and pink modules are downregulated in AD temporal cortex data[28] (Fig. 7i–k) as well as frontal cortex[23] (Fig. 7l–n). Together, these results support concordance between changes in gene expression involved in neurotransmitter-mediated signaling, mitochondrial energetics and metabolism in hAβ-KI mice and those observed in human AD cortical samples.

## Discussion

This study demonstrates that substitution of mouse Aβ with the wild-type human isoform is sufficient to produce significant changes in cognition, synaptic plasticity, inflammation, astrocyte-associated OC+/PAS granule formation, and gene expression. These physiological alterations were associated with an age-dependent increase of insoluble Aβ. The Aβ encoding exon in these hAβ-KI mice was floxed to permit temporal and cell-specific control of Aβ production. Using this feature, we showed it is possible to halt Aβ production, reverse the cognitive deficits in the advanced-aged hAβ-KI mice and reduce the quantity of OC+/PAS granules.

The discovery of AD-associated genes has provided a crucial tool for the development of AD animal models, facilitating dissection of the underlying molecular mechanisms and the identification of potential therapies[1–4,7,29,30]. Although the current rodent models have helped to advance the field, the initial optimism for developing effective treatments to slow, halt or reverse AD has been dampened by the lack of success in translating interventions into the clinic[5–7]. Many reasons may underlie the failure to translate therapies from the bench to the bedside, but one likely and significant factor is that all of the extant AD models harbor autosomal-dominant AD genes, whereas most patients do not. Importantly, because these autosomal-dominant mutations result in early onset of pathology, they preclude investigation of the influence of aging—the principal risk factor for sporadic AD. In addition, the majority of widely used AD animal models harbor supraphysiological expression of several human transgene (including APP and TAU), driven by ectopic transcriptional regulatory elements inserted randomly into the mouse genome[1–4]. The expression of genes via heterologous

regulatory elements can result in non-physiological expression patterns and, critically, may influence the outcome when studying the responsiveness of disease-associated genes to genetic and environmental risk factors, including aging[1–4]. Moreover, the insertion of transgenes can in some cases affect essential genes or regulatory elements, which could then produce disassociated biological functions and/or pathological phenotypes that are absent in the human cases[1–4]. A few animal models have been generated to circumvent these limitations, such as several mutant-bearing APP-KI mice[9,31,32], which are useful for investigating the pathways leading to FAD. We believe that the development of wild-type hAβ-KI mice described here will complement such models, and together will promote a better understanding of the influences that aging, the environment and other factors have on AD.

The hAβ-KI mice develop age-associated increase of insoluble Aβ, and decrease of soluble Aβ, reproducing previous observation in human AD cases and suggesting that progressive shift from soluble to insoluble Aβ pools may play a mechanistic role in the onset and progression of AD[33]. However, in our new model, increased insoluble Aβ (including Aβ40 and Aβ42) did not result in the formation of amyloid aggregates. Possible explanations for this finding include: (1) a higher threshold of Aβ is necessary to trigger Aβ pathology in mice, (2) considering that amyloid build up in the human brain takes place during a long asymptomatic phase, it is possible that more time is necessary for insoluble Aβ to form plaques, (3) Aβ42/40 ratio did not change across the lifespan of the hAβ-KI mice (4) age-associated impairment in clearance pathways is also necessary as in LOAD cases to facilitate Aβ plaque formation or (5) the influence of different risk factors (e.g., genetics, diet, environment, or co-morbidities) are required for triggering the onset and progression of the disease[34–38]. Nonetheless, the hAβ-KI mouse model contains seeding-competent aggregates, which increases the utility of this model to screen for additional genetic and environmental changes that can lead to formation of plaques.

This new hAβ-KI model develops OC+/PAS+ clusters of granules that are intimately associated with astrocytes in brain regions relevant to AD. These granules have previously been described in advanced-aged mice and in models of accelerated brain aging, and are carbohydrate-rich deposits composed mainly of break-down products from degenerating astrocytes, neurons and oligodendrocytes[39,40]. Alterations in astrocytic processes are also observed in the hAβ-KI model, raising the possibility that changes in astrocytic biology leads to the formation of OC+/PAS + granules. We also demonstrated that the presence of human Aβ accelerates the appearance and accumulation of OC+/PAS+ granules, and that this is further exacerbated by familial AD mutations. The appearance of these granules coincides with cognitive impairments in hAβ-KI mice, suggesting that the neurodegenerative process starts relatively early in this model. Of note, PAS granules have been described in human brains (originally by Purkinje, known as Corpora Amylacea (CA)), and their

presence is increased in AD cases as well as other degenerative conditions[40–42]. A recent study showed that CA accumulate at the same locations in the human brain as we describe in the hAβ-KI mice, and represent an important mechanism for waste clearance from the brain parenchyma to the cerebrospinal fluid (CSF), then to the meninges and cervical lymph nodes[42]. The build-up of CA (OC+/PAS granules) may indicate that such waste clearance mechanisms are being impaired, or overloaded or both, and could be an early stage in AD pathogenesis. Notably, recent evidence has shown that these same PAS/CA granules contain hypo-phosphorylated tau, in both 3xTg-AD and human AD hippocampi, and suggest that they represent an important clearance mechanism for tau from the brain parenchyma[40]. The findings that they are induced by aging and human Aβ may represent an important link between aging, Aβ, and tau pathologies in AD relevant brain regions.

We also demonstrated that humanizing the Aβ sequence in this mouse strain produces significant changes in cognitive and synaptic processes and that halting Aβ expression in the hAβ-KI model mitigates the cognitive impairments. Studies in multiple animal models have shown that synaptic deficits occur before amyloid plaque formation, supporting that Aβ oligomers are the main toxic species leading to synaptic dysfunction and cognitive impairments in AD (reviewed in[43,44]). Our study suggests that non-aggregated oligomeric species present in our model are responsible for the cognitive and synaptic deficits observed, and that the reduction of Aβ mitigates the cognitive deficits in the hAβ-KI mice. This is consistent with other studies, including one from our lab using genetically controllable transgenic model expressing a mutant form of APP, which demonstrated that cognitive and synaptic function can be restored upon reduction of APP/Aβ expression[45,46]. These findings show that wild-type human Aβ is sufficient to produce profound alterations in cognitive and synaptic processes in mice.

Transcriptomic analysis showed that ~15 genes were altered in old hAβ-KI mice compared to WT mice. Alteration of these genes has been also described in human cases, for example, Gabra2 encodes for the alpha 2 subunit of the GABA receptor (GABAα2), and a recent study in postmortem LOAD cases has shown significant alterations in the GABA receptor expression and currents, including GABAα2[47], indicating that alterations in the GABA system might be an important contributing factor to the cognitive and synaptic changes observed in AD and in the hAβ-KI mice. Mitochondrial perturbation has also been identified in sporadic AD cases, and elevated level of mitochondrial unfolded protein response (mtUPR) has been expressed as consequence of the accumulation of damaged or unfolded proteins in the mitochondria[48,49]. In this regard, increases in Hspe1, which encodes for heat shock protein family E member 1 (HSPE1), may be indicative of mitochondrial proteostasis alterations in the hAβ-KI mice and may contribute to the formation of PAS granules. Furthermore, expression of Sdhd, a nuclear gene encoding a subunit of succinate dehydrogenase (complex II) of the electron transport chain was reduced in 22-month-old hAβ-KI mice versus age-matched wild type mice. Together, these findings at the mRNA level are consistent with studies in postmortem AD cases identifying that these markers are significantly altered in humans[48–50]. Furthermore, WGCNA assay demonstrated that changes in gene expression involved in neurotransmitter-mediated signaling, mitochondrial energetics and metabolism in hAβ-KI mice are preserved in AD. Collectively, we have identified several important phenotypic features (including, synaptic, cognitive, inflammatory, neurodegenerative and transcriptomic changes) that resemble the human condition. Thus, the hAβ-KI line may prove useful for investigating the primary risk factor for AD—aging— as well as the effect of multiple risk factors (e.g., genetics, diet, environment, or co-morbidities) in the pathogenesis of the disease.

The hAβ-KI mouse model described here is an important foundation for the recently established consortium Model Organism Development and Evaluation for Late-Onset Alzheimer's disease (MODEL-AD). This NIA-led initiative seeks to: develop the next generation of animal models based on human data; institute a standardized and rigorous process for characterization of animal models; align the pathophysiological features of AD models with corresponding stages of clinical disease using translatable biomarkers; establish guidelines for rigorous preclinical testing in animal models; and ensure rapid availability of animal models, protocols and validation data to all researchers for preclinical drug development. We believe this hAβ-KI mouse will serve as a valuable component of a platform that includes other humanized genes such as MAPT (encoding TAU) or APOE4 to produce novel and improved models to investigate AD. Moreover, the hAβ-KI mouse line will also serve as useful foundation to introduce and identify factors (namely genetic, environmental and co-morbid conditions) that drive AD pathology.

## Methods

**Mouse housing.** Mice were housed in Super Mouse 750[TM] ventilated cages (Lab Products, Inc, Seaford, DE, USA). Cages contained 1/8" corn cob bedding (bed-o-cobs) and also contained 2 cotton nestlet squares for bedding. Lights were on a 12 h ON/OFF cycle, room temperature was set at 72°F with a variance of ±2°F, and ambient humidity conditions. A standard food diet (Envigo, Placentia, CA, USA; Ref 2020x) and water (purified by reverse osmosis) were provided ad libitum.

**Production of a humanized wild-type Aβ knock-in mice.** The mouse BAC RP23-424A20, containing the entire App gene, was used to amplify and sub-clone a 3.6 kb Kpn I – Pac I region containing App exon 14 (ENSMUSE00000131684; App-201 isoform ensembl.org). The Aβ coding sequence within exon 14 was modified by changing three bases to produce three amino acid substitutions (amino acids 5 (G-> R), 10 (F-> Y) and 13 (R-> H)). LoxP sites were added to flank exon 14 and a FRT site-flanked neo selection cassette (pM-30) was used to enable G418-selection in mouse ES cells. The targeting vector was electroporated into C57BL/6N derived JM8.N4 ES cells. Correctly targeted ES cells were injected into C57BL/6J blastocysts. To remove the neo selection cassette, the resulting male chimeras were bred with Actin-FLPe transgenic female mice, N5 on C57BL/6NTac (JAX Stock No. 005703). Resulting male offspring that no longer carried the selection cassette were then bred with C57BL/6J females to remove the Tg(ACTFLPe) 9205Dym transgene. The mice were then intercrossed to produce homozygotes that were subsequently intercrossed for 7–8 generations, and then bred with C57BL/6NTac wild-type mice once. Heterozygotes were intercrossed to generate homozygotes for the hAβ-KI allele that are also homozygous wild-type for Nnt. Male and female homozygous and heterozygous hAβ-KI and wild type (WT) mice aged 2-, 6-, 10-, 14-, 18- and 22-months (n = 12–19/genotype/age) were used in the current study. The hAβ-KI mice (official strain name B6(SJL)-App[tm1.1Aduci]/J) are available from The Jackson Laboratory (JAX Stock No. 030898). Homozygous hAβ-KI were crossed with homozygous B6.129-Psen1[tm1Mpm]/J mice (JAX Stock No. 004193), the resulting heterozygous hAβ-KI/ PS1[M146V] mice were intercrossed to obtain homozygous knock-in mice hAβ-KI/ PS1[M146V] and aged to 18 months. In addition, B6;129-Tg(APPSwe,tauP301L)1Lfa Psen1[tm1Mpm]/Mmjax mice (JAX Stock No. 34830) mice were used at 12, 18 and 22 month of age as positive control in many experimental readouts. All animal procedures were performed in accordance with National Institutes of Health and University of California, Irvine Animal Care and Use Committee, who approved the study protocol. Primer pairs for animal genotyping is included in Supplementary Table. 1.

**Tamoxifen administration.** B6.Cg-Ndor1 [Tg(UBC-cre/ERT2)1Ejb]/2J (JAX Stock No. 008085) were crossed with homozygous hAβ-KI mice to generate animals with which to verify that the loxP sites were functional. F1 App[hAb-KI/+]; UBC-Cre-ERT2 hemizygous mice at 4 months of age received one intraperitoneal (i.p) injection of Tamoxifen (T5648; Sigma-Aldrich, Saint Louis, MO, USA) per day during 5 days (200 μl per injection at 20 mg/ml of tamoxifen diluted in sunflower oil). Mice were sacrificed at two time points, 24 h and 1 month after the last injection. Brains were collected and immunoblotting analysis was performed as described below. In addition, hAβ-KI[+/+]/ UBC-CRE[ERT2+/−] mice were treated for 5 consecutive days with tamoxifen (5 mg per 25 g body weight) or vehicle (sunflower oil) via oral administration at 2 months of age. Mice were aged and sacrificed at 8 months of age. Primer pairs for animal genotyping is included in Supplementary Table. 1.

**Behavior analysis**. Novel object recognition: Before testing each mouse was habituated to an empty Plexiglas arena ($45 \times 25 \times 20$ cm) for 4 consecutive days. The lighting intensity in each behavioral task was measured at 44 lux. The arena and the stimulus objects were cleaned thoroughly between trials to ensure the absence of olfactory cues. If an animal did not explore both objects during the training phase for any behavioral test, the test was not scored. Animal exploration was considered if the mouse's head was within 2.54 cm (i.e., one inch) of the object, with its neck extended and vibrissae moving. Simple proximity, chewing, or standing on the object was not scored as exploratory. All exploratory segments and tests were videotaped for scoring purposes. For the novel object recognition testing, mice were exposed to two identical objects placed at opposite ends of the arena for 10 min. After 24 h, mice were presented for 10 min with one of the familiar and a novel object of similar dimensions. The discrimination index represents the percentage of time that mice spent exploring the novel object[51].

Novel object location: This behavior test was performed 1 month after virus administration[52]. Before testing each mouse was habituated to an empty white acrylic arena ($30.5 \times 30.5 \times 30.5$ cm) for 6 consecutive days. The lighting intensity in each behavioral task was measured at 44 lux. A black vertical stripe was fixed on one of the walls and arena was covered with Sani-Chips bedding (P.J Murphy Forest Products). The arena and the stimulus objects were cleaned thoroughly between trials to ensure the absence of olfactory cues. If an animal did not explore both objects during the training phase for any behavioral test, the test was not scored. Animal exploration was considered if the mouse's head was within 2.54 cm (i.e., one inch) of the object, with its neck extended and vibrissae moving. Simple proximity, chewing, or standing on the object did not count as exploratory. All exploratory segments and tests were videotaped for scoring purposes. For this test, mice were exposed to two identical objects in the arena for 10 min. After 24 h, one object was moved to a new location within the arena and mice were allowed to explore both objects for 5 min. The discrimination index represents the percentage of time that mice spent exploring the novel location.

Contextual fear conditioning: During training, mice were placed in the fear-conditioning chamber (San Diego Instruments, San Diego, CA) and allowed to explore for 2 min 30 s before receiving one electric foot shock (duration, 2 s; intensity, 0.2 mA). Animals were returned to the home cage 30 s after the foot shock. Twenty-four hours later, behavior in the conditioning chamber was video recorded during 5 min and subsequently analyzed manually for freezing by a blind investigator. Freezing was defined as the absence of all movement except for respiration.

**Tissue preparation**. After deep anesthesia with $CO_2$, WT, hAβ-KI, hAβ-KI/ PS1$^{M146V}$, hAβ-KI$^{+/+}$/ UBC-CRE$^{ERT2+/-}$ and 3xTg-AD mice were perfused transcardially with 0.1M phosphate-buffered saline (PBS, pH 7.4). Following dissection, half of the brain tissue was used for biochemical analysis and the other half for immunohistochemical analysis[51].

**Immunoblotting**. Equal amounts of protein (10–20 μg) were separated through 10% Bis-Tris gel (Invitrogen, Carlsbad, CA, USA) and transferred to nitrocellulose membranes. Membranes were blocked for 1 h in 5% (w/v) suspension of bovine serum albumin (BSA; Gemini Bio-Products, West Sacramento, CA, USA) in 0.2% Tween 20-Tris-buffered saline (TBS) (pH 7.5). After blocking, the membranes were incubated overnight at 4 °C, with one of the following primary antibodies: 6E10 (1:1000; BioLegend, San Diego, CA, USA; Catalog # 83001, Lot #B2261151), anti-APP-CT20 (1:1000; EMD Millipore, Burlington, MA, USA; Catalog #171610, Lot #D00080225), anti-APP-22C11 (1:1000; EMD Millipore, Burlington, MA, USA; Catalog #MAB348, Lot #2280425). Membranes were washed in 0.2% Tween 20-TBS for 20 min and incubated at 20 °C with the specific secondary antibody anti-rabbit IgG (H + L), HRP conjugate (1:10000 Invitrogen, Carlsbad, CA, USA; Catalog #31460, Lot #RL240411) and anti-mouse IgG (H + L), HRP conjugate (1:10000 Invitrogen, Carlsbad, CA, USA; Catalog #31430, Lot#RJ240410) for 60 min. Blots were developed using Super Signal (ThermoFisher Scientific, Rockford, IL, USA) and signal value were measured with ImageJ (64-bit) software.

**Immunohistochemistry**. Coronal free-floating sections (40 μm) were pre-treated with 3% $H_2O_2$/3% methanol in TBS for 30 min to block endogenous peroxide activity. After TBS wash, sections were incubated in TBS with 0.1% Triton X-100 (TBST) for 15 min, then in TBST with 2% bovine serum albumin (BSA, Sigma-Aldrich) for 30 min. Next, sections were incubated for 72 h at 4 °C, with 6E10 antibody (1:1000; BioLegend, San Diego, CA, USA; Catalog # 83001, Lot #B2261151), and 48 h at 22 °C with Aβ40 antibody (1:1000; EMD Millipore, Burlington, MA, USA; Catalog #AB5074P, Lot #3062336), Aβ42 antibody (1:1000; EMD Millipore, Burlington, MA, USA; Catalog #AB5078P, Lot #3172448), Iba1 antibody (1:1000 Fujifilm Wako Chemicals, Osaka, Japan; Catalog #019-19741, Lot #PTE0555) and GFAP antibody (1:5000 Abcam, Cambridge, MA, USA; Catalog #ab134436, Lot #GR3197612-3), synaptophysin antibody (1:1000 Abcam, Cambridge, MA, USA; Catalog #ab14692, Lot #GR66861-35), amyloid fibrils OC antibody (1:200 Millipore Sigma, Darmstadt Germany; Catalog #AB2286, Lot#3313147), and PSD95 antibody (1:1000 EMD Millipore, Burlington, MA, USA; Catalog #1596, Lot #UI287733). Sections were then incubated with biotinylated anti-rabbit (1:500 Vector lab, Burlingame, CA, USA; Catalog #BA-1000, Lot

#ZG0122) or anti-mouse (1:500 Vector lab, Burlingame, CA, USA; Catalog #BA-2000, Lot #ZA0409) in TBS + 2%BSA + 5% normal serum for 1 h at 20 °C, followed by Vector ABC kit and DAB reagents (Vector Laboratories, Burlingame, CA, USA) to visualize staining. Optical microscopy images were obtained with Nikon eclipse 80i microscope with the ACT-2U v1.21.41.176 software (Nikon European Headquarter, Burgerweeshuispad, Amsterdam, Netherlands).

For fluorescent stain, sections were incubated in secondary goat anti-rabbit Alexa-fluor 488 (Invitrogen, Carlsbad, CA, USA; Catalog #A11034, Lot #1937195) for Iba1 and synaptophysin, goat anti-chicken Alexa-fluor 555 for GFAP antibody (Invitrogen, Carlsbad, CA, USA; Catalog #A21437, Lot #1964371) and goat anti-mouse Alexa-fluor 555 for PSD95 antibody (Invitrogen, Carlsbad, CA, USA; Catalog #A11004, Lot #1218263) for 1 h at room temperature. Sections were then mounted and cover-slipped with Fluoromount-G with or without 4',6-diamidino-2phenylindole (DAPI) (Southern Biotech, Birmingham, AL, USA). Microglia and astroglia were modeled using Imaris software 9.2.1 (Bitplane Inc. Concord, MA, USA) and changes in these cells, such as cell body area, process length, and number of microglial branches were analyzed (5 sections per animal with Zeiss slidescanner Axio Z1 (v2.3) were used, $n = 5$–6 animals per genotype and age)[53]. Similarly, synaptic puncta were evaluated with Imaris software 9.2.1 (Bitplane Inc.) (three sections and two stack per section with a 63× objective and zoom factor #2 with a Z stack of 2 μm and 0.1 μm distance for each confocal plane were used in a SP8 Leica confocal microscope (Leica Microsistemas S.L.U, L'Hospitalet de Llobregar, Spain), $n = 5$ animals per genotype).

For Amylo-Glo, free-floating coronal sections were incubated for 5 min in 20 °C in 70% ethanol followed by a 2-min wash in Milli-Q water. The sections were then immersed in Amylo-Glo RTD Amyloid Plaque Stain Reagent (1:100; Biosensis, Thebarton, South Australia; Catalog #TR-300-AG) suspended in 0.9% saline solution for 10 min at 20 °C. Afterwards, the sections were rinsed in 0.9% saline solution for 5 min and rinsed briefly in Milli-Q water for 15 s before proceeding with immunohistochemistry protocols.

For Congo Red (C6277-25G; Sigma-Aldrich) or thioflavin-S (T-1892; Sigma-Aldrich) stain, coronal sections were incubated 3 min in a solution of 20% of Congo Red or 10 min in a solution with 0.5% thioflavin-S in 50° ethanol[53].

For Periodic Acid Schiff (PAS) stain, free-floating sections were mounted on microscope slides and rehydrated followed by two washes of Milli-Q water. The slides were immersed in 0.5% sodium periodate (ACROS Organics, New Jersey, USA; Cat #AC198381000) for 15 min at 20 °C. After rinsing with Milli-Q water three times, the slides were immersed in Schiff's Reagent (VWR, Radnor, PA; Cat #15204-138) for 5 min at 20 °C. Slides were rinsed with three 5-min washes in 0.4% sodium metabisulfite (MP Biomedicals, Irvine, CA; Cat #MP021914381) at 20 °C before a final rinse in running tap water for 5 min at 20 °C. Following two rinses of Milli-Q water, the slides were dehydrated before finally being cleared with Xylene and cover-slipped with DPX Mounting Media (Electron Microscopy Sciences, Hatfield, PA; Cat #13512).

**Stereological quantification**. Cell counts were performed in hippocampal CA1 region using Stereoinvestigator software (MBF Bioscience, Williston, VT, USA). 8–10 hippocampal sections per animal and stained with cresyl violet ($n = 5$–6 per genotype) were used through the entire antero-posterior extent of the hippocampus (between −1.46 mm anterior and −3.40 mm posterior to Bregma according to the atlas of Franklin and Paxinos, Third Edition, 2007). The stratum pyramidal of CA1 region was defined using a 5× objective and neuronal cells were counted using a 100×/1.4NA objective. The number of counting frames varied with the hippocampal region. We used a counting frame area of 2500 μm with step lengths of 200 μm × 200 μm and dissector height of 13 μm. Hippocampal volume was calculated following Cavalieri principal. 8–10 hippocampal sections per animal ($n = 5$–6 per genotype) and stained with cresyl violet. The hippocampus was defined with 5× objective and a grid size of 250 μm was used to estimate the hippocampal volume.

For cell count and volume, the coefficient of error (CE) value for each individual animal ranged between 0.03 and 0.08 for cell count and estimated volume.

**OC+ cluster quantification**. Images ($n = 5$–10 per genotype per timepoint) of mouse hippocampal regions (between −2.05 mm posterior and −3.28 mm posterior to Bregma according to the Allen Mouse Brain Atlas, Reference Atlas version 1, 2008) were obtained from the confocal microscopy program after immuno-fluorescent staining with GFAP and OC antibodies of one brain section per animal. The obtained images were then utilized to acquire individual OC+ granule count using the Imaris software 9.2.1 (Bitplane Inc.). Individual OC+ granules from OC + clusters were derived through a double-blinded study measuring the surface area (μm²), the total number of granules and the intensity per granule within the cluster area analyzed. The signal of the individual granules inside of the cluster selected was recognized and kept within a threshold mask that was kept consistent throughout the entire analysis. The area and granule count were automatically generated by the Imaris program.

**Electrochemiluminescence-linked immunoassay**. Aβ levels were quantified using the Meso Scale Discovery (MSD) 96-well multi-spot 6E10 according to the manufacturer's instruction (Meso Scale Discovery, Rockville, MD, USA)[54]. Protein extracts were prepared by homogenizing the hippocampus samples in T-per

(Thermo Fisher Scientific, Rockford, IL, USA) extraction buffer (150 mg/mL), complemented with proteases inhibitor (Complete Mini Protease Inhibitor Tables, Roche Diagnostics GmbH, Germany) and phosphatases inhibitor (5 mmol/L, Sigma Aldrich, St. Louis, MO, USA), followed by centrifugation at $100,000 \times g$ for 1 h. After that, the pellet was resuspended with 70% formic acid followed by centrifugation at $100,000 \times g$ for 1 h. T-Per soluble fractions were loaded directly onto the electrochemiluminescence-linked immunoassay plate, and the formic acid supernatants (insoluble fractions) were diluted 1:10 in neutralization buffer (1 M Tris-base and 0.5 M $NaH_2PO_4$) before loading. Standards (including $A\beta_{1-40}$, and $A\beta_{1-42}$), and samples, were added to the 96-well plate and incubated overnight, washed, and read in a Sector Imager plate reader MSD MESO QuickPlex SQ 120 (Meso Scale Discovery), immediately after addition of the MSD read buffer. $A\beta$ concentration was calculated with reference to the standard curves and expressed as micrograms per milligrams of proteins. V-Plex proinflammatory panel 1 (mouse) kit for IFN-γ, IL-1β, IL-2, IL-4, IL-10, TNF-α was used. Fifty μl of soluble fractions from hippocampal sample were added per well, calibrator and control were added in a plate and were incubated overnight, washed and read in a Sector Imager plate reader MSD MESO QuickPlex SQ 120, immediately after addition of the MSD read buffer. Cytokines levels were calculated with reference to the standard curves and expressed as micrograms per milligram of protein.

**Hippocampal slice preparation and recording.** Hippocampal slices ($n = 6$–10 sections) were prepared from male and female hAβ-KI and WT mice (2 and 18 months of age)[55]. Following isoflurane anesthesia, mice were decapitated and the brain was quickly removed and submerged in ice-cold, oxygenated dissection medium containing 124 mM NaCl, 3 mM KCl, 1.25 mM $KH_2PO_4$, 5 mM $MgSO_4$, 26 mM $NaHCO_3$, and 10 glucose without $CaCl_2$. Transverse hippocampal slices (320 μm) through the mid-third of the septotemporal axis of the hippocampus were prepared using a Leica vibrating tissue slicer (Model: VT1000S) before being transferred to an interface recording containing preheated artificial cerebrospinal fluid (aCSF) composed of 124 mM NaCl, 3 mM KCl, 1.25 mM $KH_2PO_4$, 1.5 mM $MgSO_4$, 2.5 mM $CaCl_2$, 26 mM $NaHCO_3$, and 10 mM glucose and maintained at $31 \pm 1$ °C. Slices were continuously perfused with this solution at a rate of 1.75–2 ml/min while the surface of the slices were exposed to warm, humidified 95% $O_2$/ 5% $CO_2$. Recordings began following at least 2 h of incubation.

Field excitatory postsynaptic potentials (fEPSPs) were recorded from CA1b stratum radiatum using a single glass pipette filled with 2 M NaCl (2–3 MΩ) in response to orthodromic stimulation (twisted nichrome wire, 65 μm diameter) of Schaffer collateral-commissural projections in CA1 stratum radiatum. In some slices two stimulation electrodes were used (positioned at sites CA1a and CA1c) to stimulate independent populations of synapses on CA1b pyramidal cells. Pulses were administered in an alternating fashion to the two electrodes at 0.05 Hz using a current that elicited a 50% maximal response. After establishing a 10–20 min stable baseline, one of the pathways was used to induce long-term potentiation (LTP) by delivering 5 'theta' bursts, with each burst consisting of four pulses at 100 Hz and the bursts themselves separated by 200 ms (i.e., theta burst stimulation or TBS). The stimulation intensity was not increased during TBS. The control pathway was used to monitor for baseline drifts in the slice. Data were collected and digitized for storage by NAC 2.0 Neurodata Acquisition System (Theta Burst Corp., Irvine, CA).

Data in the text are presented as means ± SD, and the fEPSP slope was measured at 10–90% fall of the slope. Data in figures on LTP were normalized to the last 10 min of baseline. Electrophysiological measures including paired-pulse facilitation, input/output curves, percent change in area of theta bursts, and LTP were analyzed using a 2-way ANOVA.

**Adeno-Associated Viruses (AVV) construct delivery.** Mouse surgery and AAV delivery were performed in 24-month-old mice. The mice were anesthetized with isoflurane (induced, 4%; maintained 1.5–2.0%) and placed in the stereotax[54]. Injection needles were lowered to the dorsal hippocampus at a rate of 0.2 mm/15 s (AP, −2.0 mm; ML, ±1.5 mm, DV, −1.5 mm relative to Bregma). Two min after reaching the target depth, 1.0 μl of virus was infused bilaterally into the dorsal hippocampus at a rate of 6 μl/h. Injection needles remained in place for two min post-injection to allow the virus to diffuse. The injectors were then raised 0.1 mm and allowed to sit for another minute before being removed at a rate of 0.1 mm per 15 s. Viral infusions were performed 1 month before behavioral analysis. Mice were injected with AAV-CAMKII-Cre for recombination. Control conditions consisted of injecting either AAV-CAMKII-GFP (no-Cre) or AAV-CAMKII-Empty vector. For all injection experiments animals were randomly assigned to the different injection conditions.

**RNA-sequencing.** Total RNAs from the hippocampus were extracted by using RNeasy Mini Kit (Qiagen, Germantown, MD). RNA integrity number (RIN) was measured by Agilent 2100 Bioanalyzer and samples with RIN ≥ 8.0 were used for cDNA synthesis and amplification followed by Smart-seq2 standard protocol. Libraries were constructed by using the Nextera DNA Sample Preparation Kit (Illumina, San Diego, CA). Libraries were base-pair selected based on Agilent 2100 Bioanalyzer profiles and normalized determined by KAPA Library Quantification Kit (Illumina). The libraries were sequenced using paired-end 43 bp mode on

Illumina NextSeq500 platform with around 10 million reads per sample. Fastq files and processed data matrices were deposited in GEO with the accession ID G9.

**Read alignment and expression quantification.** Pair-end RNA-seq reads were aligned using STAR v.2.5.1b[56] with parameters '-outFilterMismatchNmax 10 -outFilterMismatchNoverReadLmax 0.07 -outFilterMultimapNmax 10' to the reference genome GRCm38/mm10. We used STAR to then convert to transcriptome-based mapping with gene annotation Gencode v.M8. Gene expression was measured using RSEM v.1.2.22[57] with expression values normalized into transcripts per million (TPM).

**Differential expression analysis.** Libraries with uniquely mapping percentages higher than 75% were considered to be of good quality and used for downstream analysis. Protein coding and long non-coding RNA genes, with expression TPM ≥ 1 in at least two samples, were collected for subsequent analysis. Differential expression analysis was performed by using edgeR v.3.2.2[58], negative binomial distribution test was used to compute the p value for each gene (two-sided). The p value were adjusted by using the Benjamini–Hochberg (BH) methods and recorded as "FDR" for each gene. Differentially expressed genes were selected by using false discovery rate (FDR) < 0.1 and p value < 0.05.

**Gene ontology and pathway analysis.** Differentially expressed genes were analyzed for Gene ontology (GO) enrichment by Metascape 3.0[59] using a hypergeometric test corrected p value lower than 0.05. They were also analyzed for KEGG pathway enrichments using PaintOmics 3[60] with significant pathways selected based on Fisher's exact test p value < 0.05.

**Co-expression network analysis and module preservation.** Network analysis was performed using weighted gene co-expression analysis (WGCNA) (v1.69) package in R(v3.6.1)[20]. A signed network analysis approach was employed by calculating component-wise values for topological overlap for individual brain banks. First, bi-weighted mid-correlations were calculated for all pairs of genes, and then a signed similarity matrix was created. In the signed network, the similarity between genes reflects the sign of the correlation of their expression profiles. The signed similarity matrix was then raised to power β (β = 14) to emphasize strong correlations and reduce the emphasis of weak correlations on an exponential scale. The resulting adjacency matrix was then transformed into a topological overlap matrix as described[22]. Modules were defined using specific module cutting parameters which included minimum module size of 100 genes, deepSplit = 4 and threshold of correlation = 0.2. Modules with correlation greater than 0.8 were merged together. We used first principal component of the module, called module eigengene, to correlate with diagnosis and other variables. Hub genes were defined using intra-modular connectivity (kME) parameter of the WGCNA package. Gene-set enrichment analysis was done using enrich R (v2.1) package[61]. To understand the functional relationship of modules and its network properties in orthogonal datasets, we performed module preservation analysis. Module definitions from the WGCNA analysis on hAβ-KI mice was used and Z-statistics were calculated using the module preservation function in WGCNA package in R(v3.6.1).

**RNA isolation and qPCR.** Total RNA was isolated from cortical tissue using Direct-zol RNA Miniprep according to manufacturer's protocol (Zymo Research Corp, Irvine, CA). Total RNA concentration was determined using a spectro-photometer (NanoDrop Lite). cDNA was produced from 500 ng RNA using iScript cDNA synthesis kit following manufacturer's protocol (Biorad, Hercules, CA). Quantitative PCR was performed using a CFX Connect Real-Time System and iTaq SYBR mix (Bio-Rad) with the following primers: *App* forward 5'-TCCGTG TGATCTACGAGCGCAT-3', *App* reverse 5'-GCCAAGACATCGTCGGAG TAGT-3' (Origene NM_007471). *Gapdh* forward 5'-AACTTTGGCATTGTGGA AGG-3' and *Gapdh* reverse 5'-ACACATTGGGGGTAGGAACA-3' (Supplementary Table 2). Cycling for PCR amplification was as follows: enzyme activation at 95 °C for 30 sec, followed by 40 cycles at 95 °C for 5 s and at 60 °C for 30 s. Data was normalized to *Gapdh* and analyzed using the $2^{-\Delta\Delta CT}$ method. Data was collected and analyzed using Bio-Rad CFX Manager 3.1 (Biorad).

**Cyclic amplification of Aβ misfolding assay (Aβ-PMCA).** To prepare mouse brain homogenate, the hippocampal area of 18–21-month-old hAβ-KI, 3xTg-AD and WT mice ($n = 4$) was dissected and homogenized at 10% (w/v) in 1X PBS containing a cocktail of protease inhibitors (Roche Diagnostics, Indianapolis, IN). Seed-free Aβ (1-40) peptide was prepared in 10 mM NaOH and filtered through a 30-kDa cutoff filter. To perform Aβ-PMCA assay, a 200 μl of seed-free Aβ (2 μM) in aggregation buffer (0.1 M Tris-Cl, pH 7.4) containing thioflavin-T (5 μM) was placed in opaque 96-well plate and incubated in the presence of 0.001% of hAβ-KI, 3xTg-AD or WT mouse brain homogenate[62]. The reaction was subjected to cyclic agitation (1 min at 500 rpm followed by 29 min without shaking) at 20 °C. The increase in fluorescence was monitored at excitation of 435 nm and emission of 485 nm periodically, using a microplate spectrofluorometer Gemini-XS and Soft-Max Pro 5.4.5 software (Molecular Devices, Sunnyvale, CA). Samples from 4 different animals per mouse line were tested in duplicate. Differences in the kinetic of

aggregation between samples were evaluated by the estimation of the T50 kinetic parameter, which corresponds to the time needed to reach 50% of the maximum aggregation. The significance was analyzed by one-way ANOVA followed by the Tukey's multiple comparison post-Hoc tests using Prism 8.0 software (Graphpad Software, San Diego, CA).

**Statistics and reproducibility**. Unless stated, comparison between two groups was performed using Student's *t*test. For multiple comparisons, one-way or two-way analysis of variance (ANOVA) followed by Tukey's comparisons was applied using Prism 8 software (Graphpad Software Inc., San Diego, CA, USA). The significance was set at 95% of confidence. Data are presented as mean values ± SEM.

Figure 1d has been reproduced twice with similar results. Figure 2d–f has been reproduced in 6 animals per group and 10 sections per animal. Figure 3a, d has been reproduced in 5–10 animals per group and 1 section per animal and 6 stacks per section using a 20x objective with a Z-stack of 60 μm and 10 μm distance for each confocal plane. With a 63× objective, a Z-stack of 9 μm and 1.5 μm distance for each confocal plane was also used. Figure 3f has been reproduced in 7–8 animals per group and 1 section per animal and 6 stacks per section using a 20x objective with a Z-stack of 60 μm and 10 μm distance for each confocal plane. With a 63× objective, a Z-stack of 9 μm and 1.5 μm distance for each confocal plane were also used. Figure 4c has been reproduced in 5–10 animals per group and 1 section per animal and 6 stacks per section using a 20x objective with a Z-stack of 60 μm and 10 μm distance for each confocal plane. With a 63× objective, a Z-stack of 9 μm and 1.5 μm distance for each confocal plane were also used.

Figure 5d has been reproduce in 5 animals per group and 3 sections and two stack per section with a 63× objective and zoom factor #2 with a Z stack of 2 μm and 0.1 μm distance for each confocal plane were used. Similar pattern has been observed in the Fig. 5h with 3 mice treated with the AAV-CAMKII-GFP construct. Supplementary Fig. 1A has been replicated in 12 mice obtaining the same result. Supplementary Fig. 3A has been reproduced in 5–10 animals per group and 1 section per animal and 6 stacks per section using a 20× objective with a Z-stack of 60 μm and 10 μm distance for each confocal plane with a 63× objective, a Z-stack of 9 μm and 1.5 μm distance for each confocal plane was also used. Supplementary Fig. 3B has been reproduce in 2 animals and 2 sections per animal. Supplementary Fig. 3C has been reproduced in 10–11 animals per group and 1 section per animal. Supplementary Fig. 4 has been replicated in 5–6 mice. Supplementary Fig. 4C, E has been replicated in 5–6 animals per group and 5 sections per animal.

**Reporting summary**. Further information on research design is available in the Nature Research Reporting Summary linked to this article.

## Data availability
The Fastq files and processed data matrices were deposited in GEO with the accession ID GSE116199, also the RNA-seq files are available via the AD Knowledge Portal (https://adknowledgeportal.org). The AD Knowledge Portal is a platform for accessing data, analyses, and tools generated by the Accelerating Medicines Partnership (AMP-AD) Target Discovery Program and other National Institute on Aging (NIA)-supported programs to enable open-science practices and accelerate translational learning. The data, analyses and tools are shared early in the research cycle without a publication embargo on secondary use. Data is available for general research use according to the following requirements for data access and data attribution (https://adknowledgeportal.org/DataAccess/Instructions).

For access to content described in this manuscript see: https://doi.org/10.7303/syn24875599.

Full-length western blots are included in Supplementary Fig. 8. The other datasets generated during the current study are available from the corresponding author upon request since most of the results have been performed at UCI. Source data are provided with this paper.

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

## Acknowledgements

This study was initiated by the generosity of Harry Bubb through Cure Alzheimer's Fund CAF-50997 (F.M.L.). Additional support was from Alzheimer's Association NIRG-15-363477 (D.B.V.), AARF-16-440760 (S.F.) and NIRG-394284 (I.M.G.), The Larry Hillblom Foundation 2013-A-016-FEL (D.B.V.) and 2016-A-016-FEL (A.C.M.), the National Institute of Health (NIH) NIH/NIA/NINDS AG027544 (F.M.L.), AG00538 (F.M.L.), AG54884 (F.M.L.), OD010420 (F.M.L.), U54 AG054349 (F.M.L., A.J.T.), AG049562 (C.S.) NS083801 (K.N.G.) and AG056768 (K.N.G.), BrightFocus Foundation grant A2015535S (F.M.L.), by Minister of Science and Innovation grant PID2019-108911RA-100 (D.B.V.), Beatriz Galindo program BAGAL18/00052 (D.B.V.) and Institute of Health Carlos III (ISCiii) grant PI18/01557 (A.G.) co-financed by FEDER funds from European Union, by American federation of aging research-AFAR young investigator award and UC Irvine startup funds (V.S.) and UCI MIND pilot project (D.B.V.). The UCI-ADRC is funded by NIH/NIA Grant P50 AG16573 (F.M.L.). Genetically modified hAβ-KI mice were generated by the UCI Transgenic Mouse Facility, a shared resource funded in part by the Chao Family Comprehensive Cancer Center Support Grant (P30CA062203) from the National Cancer Institute. We thank Drs. Malcolm Leissring and Rodrigo Medeiros for critically reading the manuscript.

## Author contributions

Conceived and designed the experiments: D.B.V., V.S., A.G., G.R.M., K.N.G., M.A.W., A.M., A.J.T., F.M.L. Performed the experiments: D.B.V., L.C., A.C.M., L.T.E., S.F., V.S., M.M.T.N., K.D.H., D.I.J., K.M.T., J.P., S.J., E.A.K., C.N.D., G.B.G., F.G., J.C., C.J.R.O., J.A.G.L., M.S., D.P.M., X.M., C.D.C. Analysis of the data: D.B.V., V.S., S.J., E.A.K., C.N.D., I.M.G., G.R.M. Contributed to the writing and editing of the manuscript: D.B.V., S.F., A.C.M., L.T.E., V.S., S.J., E.A.K., G.B.G., F.G., C.J.R.O., M.K., K.W., R.A., C.S., A.G., I.M.G., G.R.M., M.A.W., A.M., A.J.T., K.N.G., F.M.L.

## Competing interests

The authors declare no competing interests.
