## [Peer Review File · Nature Communications]

Reviewers' comments:

Reviewer #1 (Remarks to the Author):

In this manuscript, Dr. LaFerla and colleagues reported development of a knock-in mouse model, hA β -KI mouse, in which the murine APP gene was humanized at the A β locus, then bred to homozygosity. No autosomal-dominant mutations were introduced in this mouse line and expression of wildtype human hA β was at physiological levels under the control of the endogenous App promoter. However, the hA β -KI line does not produce robust amyloid pathology, while could produce the seeds for aggregations. Since majority of mouse models incorporate disease-causing mutations in genes associated with autosomal-dominant dementias, these hA β -KI mice could serve as a useful platform to introduce and identify factors that drive AD pathology, specifically related to the sporadic form of the disease. However, there are some points that need further investigation to support the current conclusions.

1. hA β -KI line express wildtype human hA β under the control of the mouse endogenous App promoter. However, the expression level is at the hA β at physiological levels of mouse APP, but might not be the levels of human. It is still unclear whether the expression level of A β is comparable to that of human condition. A comparative analysis of brain and plasma Abeta levels between hA β -KI line and human will be more appropriate to evaluate whether the A β level in hA β -KI line are related to human condition.

2. The the exon encoding the A β region were flanked with loxP sites (Supplementary Fig. 1A). Upon induction of Cre expression with tamoxifen, APP levels were completely suppressed in tamoxifen-treated hA β -KI/CreERT2 mice versus PBS-treated hA β -KI/CreERT2 (Supplementary Fig. 2 and 8). Why deletion of A β region completely suppressed the APP accumulation was not explained. In addition, it will interesting to know whether suppression of APP expression in hA β -KI mice display phenotypes of APP knockout mice?

3. They showed that soluble A β 40 and A β 42 levels decreased during aging, while insoluble A β levels increased with aging (Fig. 2A1-A4). How soluble and insoluble is defined and measured were not described. What is the nature of the insoluble A β ?

4. They claimed that age-dependent increase in A β deposits is observed in hA β -KI mice and recognized with 6E10 antibody at 2-, 6-, 10-, 14-, 18- and 22- months of age (Fig 2C-D). However, the image of A β deposits is not clear to indicate the location and morphology of the A β deposits. A better quality image will help clear the issue. Since no evidence of thioflavin-S or Congo red-positive deposits (Supplementary Fig. 6A, B), they are not A β plaque. Then, what is the nature of the A β deposits? Is it intracellular or extracellular accumulation? What is composition of the accumulation? Is it mixture of A β 42 or A β 40 or dominated by one species? Whether they are oligomer or higher molecular aggregation? Whether accumulation of microglia and activated astrocytes are associated with the A β deposits?

5. Their findings suggest that expression of humanized A β even at physiological levels is sufficient to trigger alterations in key biological processes, including metabolic changes, synaptic plasticity, DNA methylation and memory-related functioning. Does the changes related to the human normal condition, or AD condition? Whether deletion of the A β exon will rescue the alternations induced by the human A β ?

6. They employed the protein misfolding cyclic amplification (A β -PMCA) assay to detect the presence of seeding-competent aggregates in the brain of hA β -KI mice. However, A β -PMCA is a sensitive methods to detect A β oligomers. A more vigorous in vivo approach might be a better approach to confirm the conclusion.

7. They suggested that 10 months after seed administration from human cortical extracts, hA β -KI mice contain significant increases in A β load (~60%) in sections adjacent to the area of injection, when compared to hA β -KI-PBS mice (Fig. 3A-C). However, it is unclear from the data presented in Figure 3 to show increase of A β load in sections adjacent to the area of injection. Analysis of A β load in the non-injected hippocampus part should be present as control in this figure.

Reviewer #2 (Remarks to the Author):

This is an interesting Ms reporting a transgenic (tg) mouse model in which the rodent amyloid precursor protein (app) gene was “humanized” on its A β domain replacing the endogenous app gene in its natural locus in the mouse genome by a knock-in approach.

The resulting tg line has a moderate (referred in the paper as “physiological”) expression of the human A β peptide. The models generated diffuse amyloid accumulation at late stages (18 months onwards) and no further pathology was observed. At that late stage the authors report impairments in LTP formation and the triggering of changes in the pattern of gene expression.

There was no clear evidence of a prior A β staging of intracellular accumulation, however some minor behavioral impairments in novel object recognition and fear conditioning were reported in the absence of microscopically detectable brain A β accumulation.

At later stages, the hA β tg mice displayed a diffuse amyloid aggregation which was limited to the hippocampal formation. Unfortunately, no description is provided on the histology or biochemistry of the cerebral cortex (or amygdala, which may be relevant given the deficit shown in the fear conditioning task), even when the readers would assume that there was no detectable pathology on such relevant telencephalic region.

Further to the above, the hA β tg model appeared to facilitate the aggregation of hippocampal amyloid material following the seeding with exogenous human A β material, extracted from one AD case. That component of the investigation, as presented, is not fully convincing. The histological illustration is of poor quality with considerable non-specific reactions and showing abundant immunoperoxidase staining in blood vessels. Also, one wonders whether the proposed incremental A β -immunoreaction is of the reported magnitude when images from Fig. 2C of hippocampal tissue at 18months of age already show considerable A β accumulation, possibly higher than or equivalent to that of the illustrate “PBS” control.

Figure legends are too extensive, somewhat confusing and repetitive. Much of the necessary detail could be moved to the Methods Section.

There are over claims throughout the paper. This is more evident in the Discussion section. In particular when referencing to the translational value of the model. Since it is becoming increasingly clear that both the amyloid and the tau pathology are essential components leading to clinical presentation of AD it is hard to conceive how the present model could justify sentences such as: “These characteristics more closely mimic the human situation”, ...”an ideal model for investigating the primary risk factor for AD-aging-“, “...will serve as useful platform to introduce and identify factors that drive AD pathology”.

The title also should be more conservative removing the component of “as a basis to study sporadic Alzheimer’s disease”. The authors should rather be more factual emphasizing that a first hA β KI tg mouse model can unleash synaptic, behavioral and genetic alterations.

The paper is well written and contains extensive experimentation. The main value of the communication being the illustration of the first mouse model with KI approach for the sole expression of human A β material.

A main drawback of the submission is the poor quality of the majority of the histological illustrations. Although the authors have undoubtedly raised some issues of general interest, the manuscript could be considered for publication after major revisions, including the toning down of over claims.

Minors points to address:

- line 134-136

The authors write that “expression of humanized A β is sufficient to trigger alterations in key biological processes”. This is an overclaim considering they are referring to the RNA-seq analysis and should therefore reformulate to “expression of humanized A β is sufficient to trigger alterations in GENE

REGULATING key biological processes”.

- line 167-168

Could the authors add a reference for the statement “... successfully mimics the human case, where most individuals are unaffected or show only mild amyloid deposits”

- line 179-181

The authors state that “after seed administration from human cortical extracts, hA β -KI mice contain significant increase in A β load (60%) (...) when compared to the hA β -KI-PBS mice”.

The graph (Fig. 3.B) shows values around 40% for the PBS group and 100% for the tg group. This is not a 60% increase but a 150% increase (or a 60 point difference).

In addition, it is not clear what the “% A β load” actually reflects and how it is calculated. Percentage relative to what? Please explain in the Methods section.

- line 229-230 and Suppl Fig.2

“Mice were sacrificed at two time points, 24 hours and 1 month after the last injection”

The figure contradicts this statement and shows tamoxifen treatment for 5 days and 1 month. The reviewer assumes it should be 24h and 1 month.

- line 252

How was the freezing in the fear conditioning test scored? Manually or through a software. Please briefly elaborate.

- line 283

“Enzyme-linked immunosorbent assay (ELISA)”

The MSD assays are not ELISA, there are no enzyme in the kits. They are electrochemiluminescence-linked immunoassay. Please correct throughout the text when ELISA is mentioned.

-line 362-376 “In vivo Ab seeding”

Can the authors explain the criteria used to define a plaque? Was a size exclusion criteria applied? Given the small objective (10x) used to perform the quantification of plaques and the apparent weak staining, small plaques may have gone undetected.

- line 365

“the left non-injected hippocampus was used as internal control”

Can the authors then justify not using the contralateral side to compare A β load and evaluate seeding activity? This seems like a more logical approach that would rule out any variability in A β load between animals. Showing pictures of injected versus un-injected side would also allow the reader to appreciate actual difference in A β load following injection of human cortical brain extracts.

- line 501-503

“Cluster A: increased gene expression in 22 month old hA β -KI mice compared to 2 and 22-month-old WT mice and 2-month hA β -KI mice” “

The heatmap of genes is based on the differential expression analysis shown in panels D-G. There are no panels showing a comparison between 22mo hA β -KI mice and 22 mo WT mice, contradicting the sentence above. If the analysis was performed, there should be a fifth panel

h22mon_vs_WT22mon.

Same goes for Cluster B.

- line 545-546, line 556

There is an error in the scale bars. The scale reads in m (meters), the reviewer assumes it is μ m (micron).

- Fig.2D

How was the A β load quantified? The reviewer assumes the method is similar to what is described for the seeding experiment in Fig. 3C.

- Supplementary Fig. 5B:

The authors show absence of immunoreactivity in age-matched WT controls at different ages. For consistency and comparison with the tg animals (Fig. 2B) and the negative control sections (Supplementary Fig. 5A), the pictures should display the same anterior-posterior (AP) level of the hippocampus at the same magnification. The current pictures for the age-matched WT show hippocampal sections of much more anterior levels than the tg.

We thank the reviewers for their constructive critiques, which have helped to improve the manuscript. We have addressed the reviewers' comments in the revised version of the manuscript, as detailed below. To assist the reviewers, the changes made in the revised manuscript are highlighted in blue.

Reviewer:1

In this manuscript, Dr. LaFerla and colleagues reported development of a knock-in mouse model, hA β -KI mouse, in which the murine APP gene was humanized at the A β locus, then bred to homozygosity. No autosomal-dominant mutations were introduced in this mouse line and expression of wildtype human hA β was at physiological levels under the control of the endogenous App promoter. However, the hA β -KI line does not produce robust amyloid pathology, while could produce the seeds for aggregations. Since majority of mouse models incorporate disease-causing mutations in genes associated with autosomal-dominant dementias, these hA β -KI mice could serve as a useful platform to introduce and identify factors that drive AD pathology, specifically related to the sporadic form of the disease. However, there are some points that need further investigation to support the current conclusions.

1. hA β -KI line express wildtype human hA β under the control of the mouse endogenous App promoter. However, the expression level is at the hA β at physiological levels of mouse APP, but might not be the levels of human. It is still unclear whether the expression level of A β is comparable to that of human condition. A comparative analysis of brain and plasma A β levels between hA β -KI line and human will be more appropriate to evaluate whether the A β level in hA β -KI line are related to human condition.

We apologize for any confusion and have clarified our statement to indicate that the physiological expression is with respect to the mouse. Our intent was to highlight that most of the current mouse AD models express supraphysiological levels of several human AD-related proteins (including APP), which is not observed in human AD. As reviewer suggested, a comparative analysis between hA β -KI mice (22mo) and human patients in hippocampal brain extracts showed no differences in soluble A β levels between both groups (including A β 40 and A β 42). These data indicates that A β levels expressions are similar between our recent model and humans.

and A β 42 (B), are quantified in 22 month old hA β -KI mice and human patients. The quantification shows no differences between groups.

Amyloid levels in hA β -KI mice and human patients. A) A β 40 and A β 42 levels in hippocampal brain extract from hA β -KI homozygous mice and human patients were quantified using the MSD V-PLEX Plus A β Peptide Panel 1 (6E10) Kit and analyzed using a MSD MESO QuickPlex SQ 120 reader (n=6-7). Soluble A β 40 (A) and

2. The exon encoding the A β region were flanked with loxP sites (Supplementary Fig. 1A). Upon induction of Cre expression with tamoxifen, APP levels were completely suppressed in tamoxifen-treated hA β -KI/CreERT2 mice versus PBS-treated hA β -KI/CreERT2 (Supplementary Fig. 2 and 8). Why deletion of A β region completely suppressed the APP accumulation was not explained. In addition, it will be interesting to know whether suppression of APP expression in hA β -KI mice display phenotypes of APP knockout mice?

We thank the reviewer for this comment and for suggesting the interesting experiment to compare phenotypes in non-conditional and conditional *App* KO mice. We have added a potential explanation in the text (please see page 4) as to why levels of *App* are decreased following expression of CRE, due to removal of the A β encoding exon that results in an out-of-frame downstream mRNA and have added new data in supplementary figure 1F. We agree it would be interesting to compare our mice with (non-conditional) APP-KO mice, although this was not the focus of the current report. Because the hA β -KI line is a conditional APP-KO mice, different phenotypes observed in these compared to a non-conditional APP-KO mice might arise due to a requirement for APP during development or due to compensatory gene expression resulting from loss of APP during development, or both (e.g. please see Müller et al., Nat Reviews 2017; Nicolas and Hassan. Development 2014 and Ramaker et al., Frontiers in Mol Neurosci 2016). Future experiments will be conducted to address this issue.

3. They showed that soluble A β 40 and A β 42 levels decreased during aging, while insoluble A β levels increased with aging (Fig. 2A1-A4). How soluble and insoluble is defined and measured were not described. What is the nature of the insoluble A β ?

We apologize for the omission. This information has been added to the methods section (please see page 19)

4. They claimed that age-dependent increase in A β deposits is observed in hA β -KI mice and recognized with 6E10 antibody at 2-, 6-,10-,14-,18- and 22- months of age (Fig 2C-D). However, the image of A β deposits is not clear to indicate the location and morphology of the A β deposits. A better-quality image will help clear the issue. Since no evidence of thioflavin-S or Congo red-positive deposits (Supplementary Fig. 6A, B), they are not A β plaque. Then, what is the nature of the A β deposits? Is it intracellular or extracellular accumulation? What is composition of the accumulation? Is it mixture of A β 42 or A β 40 or dominated by one species? Whether they are oligomer or higher molecular aggregation? Whether accumulation of microglia and activated astrocytes are associated with the A β deposits?

To address the reviewer's questions, we have added new images and details about A β pathology in the hA β -KI mice (please, see supplementary figure 6). As showed in the supplementary figure 6 (A1a, A2a and A3a) intraneuronal stain with 6E10 is detected. In addition, western-blot with A β 40 and A β 42 antibodies showed different pools of A β oligomers at low and high molecular weight (please, see supplementary figure 4). Furthermore, the inflammatory response has been analyzed and the outcome has been described in page 7 and 8 and in the supplementary figure 8.

5. Their findings suggest that expression of humanized A β even at physiological levels is sufficient to trigger alterations in key biological processes, including metabolic changes, synaptic plasticity, DNA methylation and memory-related functioning. Does the changes related to the human normal condition, or AD condition? Whether deletion of the A β exon will rescue the alternations induced by the human A β ?

The reviewer raises an excellent question regarding the changes in gene expression due to humanization of Ab and whether these changes are related to normal or pathological human states. The data in Fig. 5 support that at least some of these pathways are also altered in similar manner in human AD cases. To address the reviewer's second question about whether deletion of A β exon could rescue the alternations induced by the human A β in mice, we used injection of CRE-expressing AAV into the brains of aged homozygous hA β -KI mice and analyzed the effect of blocking expression of hA β via deletion of the floxed exon encoding hA β on cognition. Consistent with previous studies using hA β immunotherapy, the results (Fig. 3)

demonstrate that reduction of A β via the viral construct mitigated the cognitive impairment in these mice. We thank the reviewer for suggesting this experiment.

6. They employed the protein misfolding cyclic amplification (A β -PMCA) assay to detect the presence of seeding-competent aggregates in the brain of hA β -KI mice. However, A β -PMCA is a sensitive method to detect A β oligomers. A more vigorous *in vivo* approach might be a better approach to confirm the conclusion.

We thank the reviewer for this suggestion. To complement the *in vitro* experiments, intracerebral infusion of cortical brain extracts from our ADRC center at Irvine demonstrated that A β load is increased in seeded hA β -KI mice compared to contralateral and PBS treated mice. Together with the *in vitro* analyses, this data supports that hA β -KI mice contain competent seeds that facilitates the formation of A β pathology (Fig. 4).

We agree with the reviewer that *in vivo* approaches to test if seeds from different sources, route of administration and also time play a critical role in the A β aggregates formed in the hA β -KI mice. This will be the subject of a follow-up study.

7. They suggested that 10 months after seed administration from human cortical extracts, hA β -KI mice contain significant increases in A β load (~60%) in sections adjacent to the area of injection, when compared to hA β -KI-PBS mice (Fig. 3A-C). However, it is unclear from the data presented in Figure 3 to show increase of A β load in sections adjacent to the area of injection. Analysis of A β load in the non-injected hippocampus part should be present as control in this figure.

To address the reviewer's request, images of the contralateral non-injected hippocampal samples have been included in the study (Fig. 4), with the results described in page 10.

Reviewer:2

This is an interesting Ms reporting a transgenic (tg) mouse model in which the rodent amyloid precursor protein (app) gene was "humanized" on its A β domain replacing the endogenous app gene in its natural locus in the mouse genome by a knock-in approach. The resulting tg line has a moderate (referred in the paper as "physiological") expression of the human A β peptide. The models generated diffuse amyloid accumulation at late stages (18 months onwards) and no further pathology was observed. At that late stage the authors report impairments in LTP formation and the triggering of changes in the pattern of gene expression. There was no clear evidence of a prior A β staging of intracellular accumulation, however some minor behavioral impairments in novel object recognition and fear conditioning were reported in the absence of microscopically detectable brain A β accumulation. At later stages, the hA β tg mice displayed a diffuse amyloid aggregation which was limited to the hippocampal formation.

1) Unfortunately, no description is provided on the histology or biochemistry of the cerebral cortex (or amygdala, which may be relevant given the deficit shown in the fear conditioning task), even when the readers would assume that there was no detectable pathology on such relevant telencephalic region.

We thank the reviewer for the suggestion. We have added representative histological photos of hA β -KI mice in different brain regions and details. (Supplemental Fig. 6).

2) Further to the above, the hA β tg model appeared to facilitate the aggregation of hippocampal

amyloid material following the seeding with exogenous human A β material, extracted from one AD case. That component of the investigation, as presented, is not fully convincing. The histological illustration is of poor quality with considerable non-specific reactions and showing abundant immunoperoxidase staining in blood vessels. Also, one wonders whether the proposed incremental A β -immunoreaction is of the reported magnitude when images from Fig. 2C of hippocampal tissue at 18 months of age already show considerable A β accumulation, possibly higher than or equivalent to that of the illustrate "PBS" control. Figure legends are too extensive, somewhat confusing and repetitive. Much of the necessary detail could be moved to the Methods Section.

As the reviewer suggests, we expect similar A β load values between mice from Fig.2C and Fig.4C1. Initially, PBS injections did not accelerate A β pathology formation compared to non-injected mice (Fig. 4C1). We have added contralateral area (Fig. 4C2) as a comparison to confirm our conclusion that human extract from AD patients accelerates A β pathology in hA β -KI mice compared to contralateral and PBS treated mice. The figure legends have been reduced following the reviewer's advice. A figure with amyloid pathology in different brain regions of hA β -KI mice has been added (Supplemental Fig. 6).

3) There are over claims throughout the paper. This is more evident in the Discussion section. In particular when referencing to the translational value of the model. Since it is becoming increasingly clear that both the amyloid and the tau pathology are essential components leading to clinical presentation of AD it is hard to conceive how the present model could justify sentences such as: "These characteristics more closely mimic the human situation", ... "an ideal model for investigating the primary risk factor for AD-aging-", "...will serve as useful platform to introduce and identify factors that drive AD pathology".

We understand the reviewer's comment and we have tempered our enthusiasm through the entire manuscript. We concur with the reviewer that TAU pathology is also an important component of a useful animal model of late onset AD. We have addressed this issue in the discussion.

4) The title also should be more conservative removing the component of "as a basis to study sporadic Alzheimer's disease". The authors should rather be more factual emphasizing that a first hA β KI tg mouse model can unleash synaptic, behavioral and genetic alterations. The paper is well written and contains extensive experimentation. The main value of the communication being the illustration of the first mouse model with KI approach for the sole expression of human A β material. A main drawback of the submission is the poor quality of the majority of the histological illustrations. Although the authors have undoubtedly raised some issues of general interest, the manuscript could be considered for publication after major revisions, including the toning down of over claims.

As the reviewer suggested we have revised the manuscript to have a more conservative tone and we have improved the quality of the images. We also appreciate reviewer's suggestion about the title of our manuscript and we have revised it.

5) line 134-136. The authors write that "expression of humanized A β is sufficient to trigger alterations in key biological processes". This is an overclaim considering they are referring to the RNA-seq analysis and should therefore reformulate to "expression of humanized A β is sufficient to trigger alterations in GENE REGULATING key biological processes".

We thank the reviewer for pointing out this limitation of the previous version of the manuscript. To address one of the questions raised by Reviewer 1, we used microinjection of AAV to block production of hA β in the brains of aged mice. This demonstrated that an additional alteration in a key biological process (i.e. cognition) in the aged mice could be reversed by removal of hA β . Hence, changes in key biological processes in hA β -KI mice are not limited to changes in gene-expression. Regardless, as requested, we have

also amended the text in lines 134-136 to reflect the context of the specific changes described (i.e. in gene expression).

6) line 167-168 Could the authors add a reference for the statement "... successfully mimics the human case, where most individuals are unaffected or show only mild amyloid deposits?"

The statement has been corrected and new references have been added (see page 12)

7) line 179-181 The authors state that "after seed administration from human cortical extracts, hA β -KI mice contain significant increase in A β load (60%) (...) when compared to the hA β -KI-PBS mice". The graph (Fig, 3.B) shows values around 40% for the PBS group and 100% for the tg group. This is not a 60% increase but a 150% increase (or a 60-point difference). In addition, it is not clear what the "% A β load" actually reflects and how it is calculated. Percentage relative to what? Please explain in the Methods section.

We apologize for any confusion we have caused. We corrected the description of the A β load quantification (see page 10) and how the percentage amyloid load was calculated has been explained in the method section (see page 24).

8) line 229-230 and Suppl Fig.2 "Mice were sacrificed at two time points, 24 hours and 1 month after the last injection" The figure contradicts this statement and shows tamoxifen treatment for 5 days and 1 month. The reviewer assumes it should be 24h and 1 month.

We thank the reviewer for pointing out this typographical error. The figure has been modified to match the text (Supplementary Fig. 1E). Mice were euthanized 24h and 1 month after the last injection.

9) line 252 How was the freezing in the fear conditioning test scored? Manually or through a software. Please briefly elaborate.

As requested by the reviewer, we have described how freezing in the fear conditioning assay was scored in the method section (manually).

10) line 283 "Enzyme-linked immunosorbent assay (ELISA)" The MSD assays are not ELISA, there are no enzyme in the kits. They are electrochemiluminescence-linked immunoassay. Please correct throughout the text when ELISA is mentioned.

We thank the reviewer for pointing out this issue which has been corrected.

11) line 362-376 "In vivo Ab seeding" Can the authors explain the criteria used to define a plaque? Was a size exclusion criterion applied? Given the small objective (10x) used to perform the quantification of plaques and the apparent weak staining, small plaques may have gone undetected.

Initially, no exclusion criterion was applied and all 6E10 positive plaques were considered. Counting frame and grid size parameters were selected to minimize the likelihood of excluding a plaque.

12) line 365: "the left non-injected hippocampus was used as internal control" Can the authors then justify not using the contralateral side to compare A β load and evaluate seeding activity? This seems like a more logical approach that would rule out any variability in A β load between animals. Showing pictures of injected versus un-injected side would also allow the reader to appreciate actual difference in A β load following injection of human cortical brain extracts.

Following the reviewer's suggestion, contralateral non-injected hippocampal samples have been included in the study (see figure 4) and results described in page 10.

13) line 501-503: "Cluster A: increased gene expression in 22-month-old hA β -KI mice compared to 2 and 22-month-old WT mice and 2-month hA β -KI mice" "The heatmap of genes is based on the differential expression analysis shown in panels D-G. There are no panels showing a comparison between 22mo hA β -KI mice and 22 mo WT mice, contradicting the sentence above. If the analysis was performed, there should be a fifth panel h22mon_vs_WT22mon. Same goes for Cluster B.

Heatmap of genes was generated following the differential expression from panel G that includes 22mo hA β -KI mice and WT mice. In the heatmap, we decided also to include the gene expression levels of 2mo hA β -KI mice and WT mice to determine how these genetic signatures change through aging in both groups of mice. A clearer description has been added to the text (see page 6).

14) line 545-546, line 556: There is an error in the scale bars. The scale reads in m (meters), the reviewer assumes it is μ m (micron).

The reviewer is correct. We apologize for the typographical error in the scale bars, which has been corrected.

15) Fig.2D: How was the A β load quantified? The reviewer assumes the method is similar to what is described for the seeding experiment in Fig. 3C.

We have updated the method used to quantify A β in the method section (see page 18).

16) Supplementary Fig. 5B: The authors show absence of immunoreactivity in age-matched WT controls at different ages. For consistency and comparison with the tg animals (Fig. 2B) and the negative control sections (Supplementary Fig. 5A), the pictures should display the same anterior-posterior (AP) level of the hippocampus at the same magnification. The current pictures for the age-matched WT show hippocampal sections of much more anterior levels than the tg.

As reviewer requested we have added new images of Tg mice and negative control sections at similar levels in supplementary figure 5.

Reviewers' comments:

Reviewer #2 (Remarks to the Author):

The authors have revised the original communications exhaustively and responded positively to nearly all the comments.

The communication is of value in that it is indeed the first humanized A β , transgenic mice. The title still remains problematic. On the grammatical contextual sense it is not logical to refer to progressive pathology as an impairment (i.e. "age related impairments in x, x and pathology")

It is also problematic to describe the model as a basis to model late onset Alzheimer's disease (AD). The model is limited to an incipient AD amyloid pathology which does not evolve into a true human-like late onset AD pathology. It is also well below displaying the characteristic of AD like tau pathology, brain atrophy, CNS inflammation, and vascular pathology just to name a few classical pathology components. The fact that some cognitive impairments are found in this model with incipient pathology is more a reflection of the rodent's lack of neural reserve than reproducing aspects of LOAD. In this regard, it is worth noting that non-cognitively humans might display much more advanced amyloid pathology.

In sum, the newly developed A β humanized model remains of value and the findings are of interest but the title should be better adjusted to match the characteristics of the transgenic models as reported here, i.e. limited amyloid pathology even at late ontological stages. Perhaps a title along the general lines of "A humanized A β -mouse transgenic model replicating the early Alzheimer's like amyloid pathology" is more subject to Editorial decision.

Reviewer #3 (Remarks to the Author):

In this ms from Baglietto-Vargas et al a new human Abeta knock-in mouse model is presented. This is an interesting approach and the model would be useful as a more relevant alternative as compared to the plethora of APP Tg mice overexpressing APP, and complementary to the previous APP knock-in mice designed with FAD-linked mutations. The mice could for example be used in further genetic manipulations to deduce factors and pathways leading to SAD. The study is well performed following the mice upon aging and characterizing them with e-phys, behavior, gene expression analysis, Abeta seeding experiments, Abeta pathology and neuroinflammation characterization. The major concern I have is about the description and characterization of this new mouse model in terms of Abeta pathology, since this is the basis and the driver of the downstream effects investigated in the mice and will be of pivotal importance to elucidate the relevance and usefulness of the new model. The Abeta pathology is rather modest (late onset, area-restricted pathology with low abeta burden, Thioflavin and congo negative), which may very well be so taking into account that the Abeta levels are physiological and that no increase in pathology-driven Abeta42/40 ratio is present.

The authors need to carefully characterize the Abeta pathology in their KI mouse model (which should also be presented in the first figure of ms) by using antibodies and immuno-reagents recognizing specifically Abeta and not Abeta/APP which is the case of 6E10. Below is a list of concerns that should be addressed.

1. Measurements of Tris-soluble and Gu-HCl-soluble abeta40 and abeta42 by abeta specific ELISA (such as those available at Wako and IBL) are required. Please also calculate the abeta42/40 ratio.
2. Immunohistochemical assessments of the Abeta plaque depositions in the brains of the mice using N- and C-terminal specific antibodies against the Abeta peptide, including C40 and C42 are needed to really dissect what abeta species are present in the plaque. Using 6E10 to investigate especially intracellular abeta will not give any useful information since both APP and Abeta will be recognized (which the authors show in e.g. Supp Fig. 2). I would recommend IF instead of DAB, take images of the whole brain section to clearly visualize the spatial distribution of the abeta plaque load, and then

provide area-specific images. Are there any pyro-Abeta present in the plaques? That could be a sign of core plaque formation.

3. The data of a potential neuroinflammation in the brains is rather modest. Have the authors looked at astrocytes and microglia activation surrounding the abeta plaques?

4. The e-phys data suggest alterations at the synaptic level. It would be interesting to look at gamma oscillations in acute slices.

5. The decreased LTP suggest changes at the synapse, how about PSD95 and synaptophysin as basic markers for decrease in synapse density.

6. Changes in dendrites, Golgi staining could reveal alterations.

7. Are there any atrophies? Most likely there are no atrophies taking the mild pathology into account.

8. For gene expression analysis, using an $FDR < 0.1$ is suitable to judge significant changes. The authors use an $FDR < 0.3$. No validation by qPCR of hits from RNA seq was performed in Fig 1.

Further comments from reviewer 3 regarding previous reviewer 1 points raised and how they were addressed in the revision.

1. Data look robust and the abeta levels as measured by MSD are in the same ball park for their mouse model compared to human AD brain. Since this assay is based on APP/Abeta antibodies I would recommend using Abeta ELISA kits with Ab40 and Ab42 c-terminal specific antibodies to confirm MSD data.

2. The APP "KO" effect of the tamoxifen-induced deletion of the abeta sequence is probably due a destabilizing effect on the App mRNA, which has observed previously when intron 16 was deleted (Saito et al, Nat Neurosci 2014). To further investigate the phenotype of tamoxifen-induced Abeta KO mice is beyond the scope of this study.

3. Details are lacking on how the different fractions were prepared. Experimental details need to added. How was the brains homogenized, buffer conditions, centrifugation speed etc

4. Further characterization of the Abeta pathology is required. 6E10 antibody can't be used to detect intracellular abeta since this antibody will bind APP, APP fragments and Abeta. Immunohistochemical staining with abeta40 and abeta42-specific antibodies are crucial. See my comments in my review report.

5. ok

6. ok

7. ok

We thank the reviewers for their constructive critiques, which have helped to markedly improve the manuscript. We also appreciate everyone's patience and understanding as the COVID-19 has drastically curtailed research activity here at UCI as it has elsewhere (we are only allowed 30% occupancy in the labs, and initially, all non-essential research was limited). We have addressed the reviewers' comments in the revised version of the manuscript, as detailed below. To assist the reviewers, the changes made in the revised manuscript are highlighted in blue.

Reviewer #2 (Remarks to the Author):

The authors have revised the original communications exhaustively and responded positively to nearly all the comments. The communication is of value in that it is indeed the first humanized A β , transgenic mice. The title still remains problematic. On the grammatical contextual sense it is not logical to refer to progressive pathology as an impairment (i.e. "age related impairments in x, x and pathology"). It is also problematic to describe the model as a basis to model late onset Alzheimer's disease (AD). The model is limited to an incipient AD amyloid pathology which does not evolve into a true human-like late onset AD pathology. It is also well below displaying the characteristic of AD like tau pathology, brain atrophy, CNS inflammation, and vascular pathology just to name a few classical pathology components. The fact that some cognitive impairments are found in this model with incipient pathology is more a reflection of the rodent's lack of neural reserve than reproducing aspects of LOAD. In this regard, it is worth noting that non-cognitively humans might display much more advanced amyloid pathology. In sum, the newly developed A β humanized model remains of value and the findings are of interest but the title should be better adjusted to match the characteristics of the transgenic models as reported here, i.e. limited amyloid pathology even at late ontological stages. Perhaps a title along the general lines of "A humanized A β -mouse transgenic model replicating the early Alzheimer's like amyloid pathology" is more subject to Editorial decision.

We have modified the title per the reviewer's suggestion and hope this is considered to be more representative. We hope this is suitable but are open to further modifications if necessary.

Reviewer #3 (Remarks to the Author):

In this ms from Baglietto-Vargas et al a new human Abeta knock-in mouse model is presented. This is an interesting approach and the model would be useful as a more relevant alternative as compared to the plethora of APP Tg mice overexpressing APP, and complementary to the previous APP knock-in mice designed with FAD-linked mutations. The mice could for example be used in further genetic manipulations to deduce factors and pathways leading to SAD. The study is well performed following the mice upon aging and characterizing them with e-phys, behavior, gene expression analysis, Abeta seeding experiments, Abeta pathology and neuroinflammation characterization. The major concern I have is about the description and characterization of this new mouse model in terms of Abeta pathology, since this is the basis and the driver of the downstream effects investigated in the mice and will be of pivotal importance to elucidate the relevance and usefulness of the new model. The Abeta pathology is rather modest (late onset, area-restricted pathology with low abeta burden, Thioflavin and congo negative), which may very well be so taking into account that the Abeta levels are physiological and that no increase in pathology-driven Abeta42/40 ratio is present. The authors need to carefully characterize the Abeta pathology in their KI mouse model (which should also be presented in the first figure of ms) by using antibodies and immuno-reagents recognizing specifically

Abeta and not Abeta/APP which is the case of 6E10. Below is a list of concerns that should be addressed.

Accordingly with the reviewer suggestions, we have reorganized the manuscript and incorporated more detail analysis in terms of amyloid pathology. Please, see new figure 2.

1. Measurements of Tris-soluble and Gu-HCl-soluble abeta40 and abeta42 by abeta specific ELISA (such as those available at Wako and IBL) are required. Please also calculate the abeta42/40 ratio.

The A β measurements were performed via A β specific ELISA assays, with the exception that instead of a regular chromogen (i.e. TMB used by Wako and IBL) the MSD platform uses chemiluminescence, which provides greater sensitivity and dynamic range (https://www.mesoscale.com/en/technical_resources/our_technology/ecl). We have added the 42/40 ratios as requested, please see figure supplementary 2.

2. Immunohistochemical assessments of the Abeta plaque depositions in the brains of the mice using N- and C-terminal specific antibodies against the Abeta peptide, including C40 and C42 are needed to really dissect what abeta species are present in the plaque. Using 6E10 to investigate especially intracellular abeta will not give any useful information since both APP and Abeta will be recognized (which the authors show in e.g. Supp Fig. 2). I would recommend IF instead of DAB, take images of the whole brain section to clearly visualize the spatial distribution of the abeta plaque load, and then provide area-specific images. Are there any pyro-Abeta present in the plaques? That could be a sign of core plaque formation.

We appreciate the feedback and have undertaken a thorough analyses of the 6E10 deposits we originally described with DAB, and found that we were unable to detect them with fluorescence. We put this down to the increased sensitivity of DAB, but upon screening with additional antibodies (in both DAB and fluorescence) we were unable to see these deposits. In addition (and in response to your comment below) no glia responses were observed, which we would expect in the presence of plaques. As such we have removed the 6E10+ deposits from the manuscript. Of note, in performing these studies we were able to detect the appearance of OC+ (an antibody used to stain Abeta plaques and fibrils) puncta in the hippocampus, and have explored these in depth to show that they are induced by human A β , and further exacerbated by FAD mutations.

3. The data of a potential neuroinflammation in the brains is rather modest. Have the authors looked at astrocytes and microglia activation surrounding the abeta plaques?

As reviewer mentioned, neuroinflammatory changes are modest, however, we do now describe sparse astrocyte associated OC+ granule clusters in the hippocampus and piriform cortex of the hA β -KI mice. Recently, these granules have been linked to a novel waste clearance from the brain, and the induction that we see with wild type human A β and FAD mutations suggests that they may play roles in AD pathogenesis.

4. The e-phys data suggest alterations at the synaptic level. It would be interesting to look at gamma oscillations in acute slices.

The reviewer raises an excellent question regarding the changes in synaptic function and further experiments in the future will be performed to determine whether gamma oscillations occur in our new line. Since this comment, gamma oscillations have been described in a different APP-KI mouse (with the presence of FAD mutations) – Jun et al., 2020, and suggests that this would be an excellent follow up line of research to utilize our hA β -KI line, and take advantage of the loxp sites.

Jun et al., 2020. Neuron. Disrupted place cell remapping and impaired grid cells in a Knockin model of Alzheimer's disease. Jul 13;S0896-6273(20) 30477-3.

5. The decreased LTP suggest changes at the synapse, how about PSD95 and synaptophysin as basic markers for decrease in synapse density.

To address the reviewer's request, histological analysis with PSD95 and synaptophysin has been performed at 18 month of age in wt and hA β -KI mice and supporting the LTP and cognitive deficits of the mice. Please see data in figure 5 and page 9.

6. Changes in dendrites, Golgi staining could reveal alterations.

We thank the reviewer for the suggestion and future studies will be directly to address further synaptic and structural alterations of dendritic spines. In the current study, we provide suggesting evidences (LTP and histological analysis) that synaptic alterations is developed in the hA β -KI mice

7. Are there any atrophies? Most likely there are no atrophies taking the mild pathology into account.

We thank the reviewer for this comment and for suggesting if this model present any atrophy. We have observed several indicative sign of neurodegenerative processes such as changes in hippocampal volume and the formation of Periodic Acid Schiff (PAS) granules, which is associated with the malfunction of waste element degradation machinery or strengthen the production of this aggregates induced by stressor factors. In the current study, we performed a deep analysis demonstrating that the humanized A β trigger the formation of this granules in our new line.

8. For gene expression analysis, using an FDR<0.1 is suitable to judge significant changes. The authors use an FDR<0.3. No validation by qPCR of hits from RNA seq was performed in Fig 1.

Following the reviewer suggestion, gene expression analysis has been re-adjusted using FDR<0.1. Some validation of our RNAseq findings have been already provided. For example no differences in APP expression is observed in our model by RNAseq and immunoblot analysis (Please, see figure 1).

Further comments from reviewer 3 regarding previous reviewer 1 points raised and how they were addressed in the revision.

1. Data look robust and the abeta levels as measured by MSD are in the same ball park for their mouse model compared to human AD brain. Since this assay is based on APP/Abeta antibodies I

would recommend using Abeta ELISA kits with Ab40 and Ab42 c-terminal specific antibodies to confirm MSD data.

We thank the reviewer for the suggestion, and as mentioned before, the A β measurements were performed via A β specific ELISA assays, with the exception of the MSD platform using chemiluminescence instead of a regular chromogen (i.e. TMB used by Wako and IBL).

2. The APP "KO" effect of the tamoxifen-induced deletion of the abeta sequence is probably due a destabilizing effect on the App mRNA, which has observed previously when intron 16 was deleted Saito et al, Nat Neurosci 2014). To further investigate the phenotype of tamoxifen-induced Abeta KO mice is beyond the scope of this study.

We totally agree with reviewer's comment, and investigating the phenotype of tamoxifen-induced Abeta KO mice is beyond the scope of the current work.

3. Details are lacking on how the different fractions were prepared. Experimental details need to added. How was the brains homogenized, buffer conditions, centrifugation speed etc

To address the reviewer's request, more detailed description has been added to the method section. Please, see page 23 of the manuscript.

4. Further characterization of the Abeta pathology is required. 6E10 antibody can't be used to detect intracellular abeta since this antibody will bind APP, APP fragments and Abeta. Immunohistochemical staining with abeta40 and abeta42-specific antibodies are crucial. See my comments in my review report.

Following reviewer's suggestion, immunohistochemical stain for A β 40 and A β 42 has been added to the manuscript. Please, see figure 2.

Reviewers' comments:

Reviewer #3 (Remarks to the Author):

In this revised version of the ms, the authors have removed the previous findings interpreted as A β plaque pathology and the authors conclude that their A β model does not exhibit robust A β plaque pathology. Instead the authors have used an anti-protofibril antibody to reveal OC pos clusters which are increased in the A β model and further exacerbated upon crossing the A β model with PS1 mice and which follows a certain spreading pattern in the brain of the mice. This indicates that some form of A β pathology is initiated. One control experiment needs to be added; to stain 3xTg samples with this protofibril antibody used in Fig 3. I would also add a GFAP western blots since there are indications of an astrocytosis around the OC clusters.

Furthermore, the authors claim altered synaptic density by IHC of synaptophysin and PSD95. This needs to be verified by western blot of synaptic fractions of the brains from wt and A β mice.

In the discussion about the reason of the absence of A β pathology, I miss a discussion around the fact that Ab42/40 ratios are not increased which would have certainly induced a stronger A β pathology.

We thank the reviewers for their constructive critiques; we have addressed their concerns in the revised version of the manuscript, as detailed below. To assist the reviewers, the changes made in the revised manuscript are highlighted in blue.

Reviewer #3 (Remarks to the Author):

In this revised version of the ms, the authors have removed the previous findings interpreted as A β plaque pathology and the authors conclude that their A β model does not exhibit robust A β plaque pathology. Instead the authors have used an anti-protofibril antibody to reveal OC pos clusters which are increased in the A β model and further exacerbated upon crossing the A β model with PS1 mice and which follows a certain spreading pattern in the brain of the mice. This indicates that some form of A β pathology is initiated. One control experiment needs to be added; to stain 3xTg samples with this protofibril antibody used in Fig 3. I would also add a GFAP western blots since there are indications of an astrocytosis around the OC clusters.

OC-immunostaining in 3xTg-AD mice has been added in supplementary figure 3B and western blot for GFAP in supplementary figure 5 E and F.

Furthermore, the authors claim altered synaptic density by IHC of synaptophysin and PSD95. This needs to be verified by western blot of synaptic fractions of the brains from wt and A β mice.

Following the reviewer's suggestion, western blot analysis of the synaptic fraction was performed. These new data show that these critical proteins, synaptophysin and PSD95, are lower in the genetically-modified mice (Supplementary figures 4 A and B).

In the discussion about the reason of the absence of A β pathology, I miss a discussion around the fact that Ab42/40 ratios are not increased which would have certainly induced a stronger A β pathology.

We have modified the text of the manuscript to indicate that discuss the Ab42/40 ratio in the Discussion section (please, see page 14).

Reviewers' comments:

Reviewer #3 (Remarks to the Author):

In this revised version of the ms the authors have performed additional experiments.

1. OC staining of 3xTg mouse brain which is ok.
2. Synaptophysin and PSD95 levels by wb in synaptic fractions. Unfortunately the quality is so low, with bands missing so a correct interpretation of the data is not possible. Either repeat or remove the data.
3. GFAP levels by wb. Unfortunately the quality is so low, with bands missing so a correct interpretation of the data is not possible. Either repeat or remove the data.

We thank the reviewer for the constructive critiques and we have addressed their last concerns in the revised version of the manuscript, as detailed below. The changes made in the revised manuscript are highlighted in yellow.

Reviewer #3 (Remarks to the Author):

1. OC staining of 3xTg mouse brain which is ok.

We are glad that the reviewer is fine with our OC stain in 3xTg-AD mice

2. Synaptophysin and PSD95 levels by wb in synaptic fractions. Unfortunately, the quality is so low, with bands missing so a correct interpretation of the data is not possible. Either repeat or remove the data.

Following reviewer's suggestion, we have removed the data.

3. GFAP levels by wb. Unfortunately, the quality is so low, with bands missing so a correct interpretation of the data is not possible. Either repeat or remove the data.

Following reviewer's suggestion, we have removed the data.